# RoDyn-SLAM: Robust Dynamic Dense RGB-D SLAM with Neural Radiance Fields

## Abstract

Leveraging neural implicit representation to conduct dense RGB-D SLAM has been studied in recent years. However, this approach relies on a static environment assumption and does not work robustly within a dynamic environment due to the inconsistent observation of geometry and photometry. To address the challenges presented in dynamic environments, we propose a novel dynamic SLAM framework with neural radiance field. Specifically, we introduce a motion mask generation method to filter out the invalid sampled rays. This design effectively fuses the optical flow mask and semantic mask to enhance the precision of motion mask. To further improve the accuracy of pose estimation, we have designed a divide-and-conquer pose optimization algorithm that distinguishes between keyframes and non-keyframes. The proposed edge warp loss can effectively enhance the geometry constraints between adjacent frames. Extensive experiments are conducted on the two challenging datasets, and the results show that RoDyn-SLAM achieves state-of-the-art performance among recent neural RGB-D methods in both accuracy and robustness.

## 1 Introduction

Dense visual simultaneous localization and mapping (SLAM) is a fundamental task in 3D computer vision and robotics, which has been widely used in various forms in fields such as service robotics, autonomous driving, and augmented/virtual reality (AR/VR). It is defined as reconstructing a dense 3D map in an unknown environment while simultaneously estimating the camera pose, which is regarded as the key to achieving autonomous navigation for robots (Leonard & Durrant-Whyte, 1991). However, the majority of methods assume a static environment, limiting the applicability of this technology to more practical scenarios. Thus, it becomes a challenging problem that how the SLAM system can mitigate the interference caused by dynamic objects.

Traditional visual SLAM methods using semantic segmentation prior (Yu et al., 2018; Bescos et al., 2018; Xiao et al., 2019; Zhang et al., 2020a), optical flow motion (Sun et al., 2018; Cheng et al., 2019; Zhang et al., 2020b) or re-sampling and residual optimization strategies (Mur-Artal & Tardós, 2017; Campos et al., 2021; Palazzolo et al., 2019) to remove the outliers under dynamic environments, which can improve the accuracy and robustness of pose estimation. However, re-sampling and optimization methods can only handle small-scale motions and often fail when encountering large-scale continuous object movements. Moreover, semantic priors are specific to particular categories and can not represent the real motion state of the observation object. The above learning-based methods often exhibit a domain gap when applied in real-world environments, leading to the introduction of prediction errors.

Recently, dense visual SLAM with neural implicit representation has gained more attention and popularity. This novel map representation is more compact, continuous, efficient, and able to be optimized with differentiable rendering, which has the potential to benefit applications like navigation, planning, and reconstruction. Moreover, the neural scene representations have attractive properties for mapping, including improving noise and outlier handling, geometry estimation capabilities for unobserved scene parts, high-fidelity reconstructions with reduced memory usage, and the ability to generate high-quality static background images from novel views. Existing methods like iMap (Sucar et al., 2021) and NICE-SLAM (Zhu et al., 2022) respectively leverage single MLP and hierarchical feature grids to achieve a consistent geometry representation. However, these methods have

limited capacity to capture intricate geometric details. Recent works such as Co-SLAM (Wang et al., 2023) and ESLAM (Johari et al., 2023) explore hash encoding or tri-plane representation strategy to enhance the capability of scene representation and the system's execution efficiency. However, all these above-mentioned methods do not perform well in dynamic scenes. The robustness of these systems significantly decreases, even leading to tracking failures when dynamic objects appear in the environment.

To tackle these problems, we propose a novel NeRF-based RGB-D SLAM that can reliably track camera motion in indoor dynamic environments. One of the key elements to improve the robustness of pose estimation is the motion mask generation algorithm that filters out the sampled rays located in invalid regions. By incrementally fusing the optical flow mask (Xu et al., 2022), the semantic segmentation mask (Jain et al., 2023) can become more precise to reflect the true motion state of objects. To further improve the accuracy of pose estimation, we design a divide-and-conquer pose optimization algorithm for keyframes and non-keyframes. While an efficient edge warp loss is used to track camera motions for all keyframes and non-keyframes w.r.t. adjacent frames, only keyframes are further jointly optimized via rendering loss in the global bundle adjustment (GBA).

In summary, our **contributions** are summarized as follows: **(i)** To the best of our knowledge, this is the first robust dynamic RGB-D SLAM with neural implicit representation. **(ii)** In response to the issue of inaccurate semantic priors, we propose a motion mask generation strategy fusing spatio-temporally consistent optical flow masks to improve the robustness of camera pose estimation and quality of static scene reconstruction. **(iii)** Instead of a single frame tracking method, we design a novel mixture pose optimization algorithm utilizing an edge warp loss to enhance the geometry consistency in the non-keyframe tracking stage. **(iv)** We evaluate our method on two challenging dynamic datasets to demonstrate the state-of-the-art performance of our method in comparison to existing NeRF-based RGB-D SLAM approaches.

## 2 RELATED WORK

**Conventional visual SLAM with dynamic objects filter** Dynamic object filtering aims to reconstruct the static scene and enhance the robustness of pose estimation. Prior methods can be categorized into two groups: the first one utilizes the re-sampling and residual optimization strategies to remove the outliers (Mur-Artal & Tardós, 2017; Campos et al., 2021; Palazzolo et al., 2019). However, these methods can only handle small-scale motions and often fail when encountering large-scale continuous object movements. The second group employs the additional prior knowledge, such as semantic segmentation prior (Yu et al., 2018; Bescos et al., 2018; Xiao et al., 2019; Bescos et al., 2021; Zhang et al., 2020a) or optical flow motion (Sun et al., 2018; Cheng et al., 2019; Zhang et al., 2020b) to remove the dynamic objects. However, all these methods often exhibit a domain gap when applied in real-world environments, leading to the introduction of prediction errors. In this paper, we propose a motion mask generation strategy that complements the semantic segmentation mask with warping optical flow masks (Teed & Deng, 2020; Xu et al., 2022), which is beneficial for reconstructing more accurate static scene maps and reducing observation error.

**RGB-D SLAM with neural implicit representation** Neural implicit scene representations, also known as neural fields (Mildenhall et al., 2021a), have garnered significant interest in RGB-D SLAM due to their expressive capacity and minimal memory requirements. iMap (Sucar et al., 2021) firstly adopts a single MLP representation to jointly optimize camera pose and implicit map throughout the tracking and mapping stages. However, it suffers from representation forgetting problems and fails to produce detailed scene geometry. DI-Fusion (Huang et al., 2021) encodes the scene prior in a latent space and optimizes a feature grid, but it leads to poor reconstruction quality replete with holes. NICE-SLAM (Zhu et al., 2022) leverages a multi-level feature grid enhancing scene representation fidelity and utilizes a local feature update strategy to reduce network forgetting. However, it remains memory-intensive and lacks real-time capability. More recently, existing methods like Vox-Fusion (Yang et al., 2022), Co-SLAM (Wang et al., 2023), and ESLAM (Johari et al., 2023) explore sparse encoding or tri-plane representation strategy to improve the quality of scene reconstruction and the system's execution efficiency. All these methods have demonstrated impressive results based on the strong assumptions of static scene conditions. The robustness of these systems significantly decreases when dynamic objects appear in the environment. Our SLAM system aims

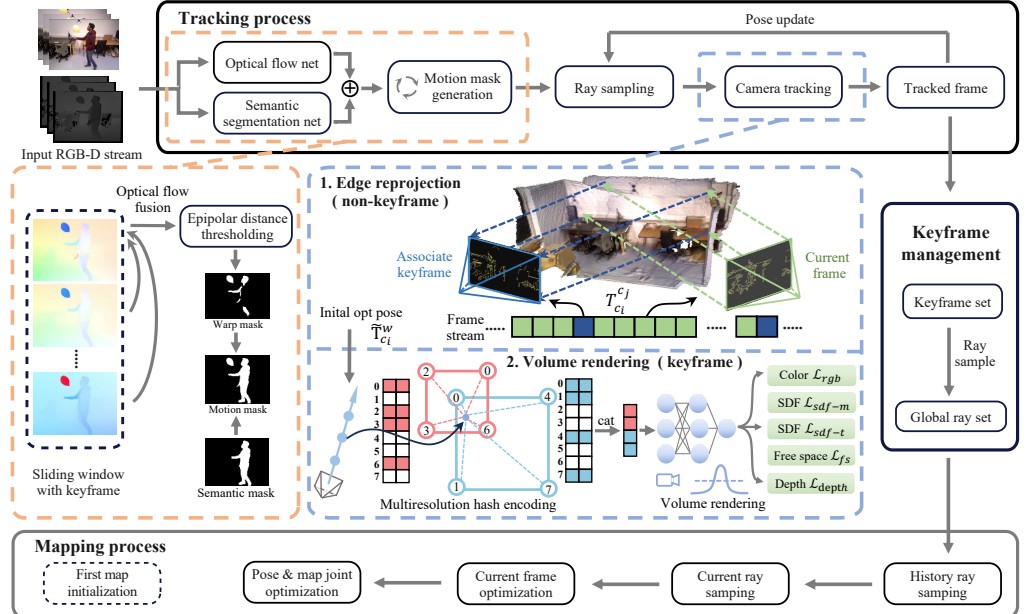

Figure 1: **The schematic illustration of the proposed method.** Given a series of RGB-D frames, we simultaneously construct the implicit map and camera pose via multi-resolution hash gird with the geometric loss $\mathcal{L}_{sdf\text{-}m}, \mathcal{L}_{sdf\text{-}t}, \mathcal{L}_{fs}, \mathcal{L}_{depth}$, color loss $\mathcal{L}_{color}$, and edge warp loss $\mathcal{L}_{edge}$.

to enhance the accuracy and robustness of pose estimation under dynamic environments, which can expand the application range for the NeRF-based RGB-D SLAM system.

**Dynamic objects decomposition in NeRFs**    As the field of NeRF continues to advance, some researchers are attempting to address the problem of novel view synthesis in the presence of dynamic objects. One kind of solution is to decompose the static background and dynamic objects with different neural radiance fields like (Martin-Brualla et al., 2021; Park et al., 2021a; Pumarola et al., 2021; Park et al., 2021b; Gao et al., 2021; Chen & Tsukada, 2022; Wu et al., 2022). The time dimension will be encoded in latent space, and novel view synthesis is conducted in canonical space. Although these space-time synthesis results are impressive, these techniques rely on precise camera pose input. Robust-Dynrf (Liu et al., 2023) jointly estimate the static and dynamic radiance fields along with the camera parameters (poses and focal length), which can achieve the unknown camera pose training. However, it can not directly apply to RGB-D SLAM system for large-scale tracking and mapping. Another kind of solution is to ignore the dynamic objects' influence by utilizing robust loss and optical flow like Chen & Tsukada (2022); Sabour et al. (2023). Compared to the dynamic NeRF problem, we often focus on the accuracy of pose estimation and the quality of static reconstruction without a long training period. Thus, we also ignore modeling dynamic objects and propose a robust loss function with a novel optimization strategy to recover the static scene map.

## 3 APPROACH

Given a sequence of RGB-D frames $\{I_i, D_i\}_{i=1}^{N}, I_i \in \mathbb{R}^3, D_i \in \mathbb{R}$, our method (Fig. 1) aims to simultaneously recover camera poses $\{\xi_i\}_{i=1}^{N}, \xi_t \in \mathbb{SE}(3)$ and reconstruct the static 3D scene map represented by neural radiance fields in dynamic environments. Similar to most modern SLAM systems (Klein & Murray, 2007; Newcombe et al., 2011), our system comprises two distinct processes: the tracking process as the frontend and the mapping process as the backend, combined with keyframe management $\{F_k\}_{k=1}^{M}$ and neural implicit map $f_\theta$. Invalid sampling rays within dynamic objects are filtered out using a motion mask generation approach. Contrary to the conventional constant-speed motion model in most systems, we introduce an edge warp loss for optimization in non-keyframes to enhance the robustness of pose estimation. Furthermore, keyframe poses and the implicit map representations are iteratively optimized using differentiable rendering.

### 3.1 IMPLICIT MAP REPRESENTATION

We introduce two components of our implicit map representation: an efficient multi-resolution hash encoding $\mathbb{V}_\alpha$ to encode the geometric information of the scene, and individual tiny MLP decoders $f_\phi$ to render the color and depth information with truncated signed distance (TSDF) prediction.

**Multi-resolution hash encoding** We use a multi-resolution hash-based feature grid $\mathbb{V}_\alpha = \{V_\alpha^l\}_{l=1}^L$ and individual shallow MLPs to represent the implicit map following Instant-NGP (Müller et al., 2022). The spatial resolution of each level is progressively set between the coarsest resolution, denoted as $R_{min}$, and the finest resolution, represented as $R_{max}$. Given a sampled point $\mathbf{x}$ in 3D space, we compute the interpolate feature $V_\alpha^l(\mathbf{x})$ from each level via trilinear interpolation. To obtain more complementary geometric information, we concat the encoding features from all levels as the input of the MLPs decoder. While simple MLPs can lead to the issue of catastrophic forgetting (Sucar et al., 2021; Zhu et al., 2022), this **mechanism of forgetfulness** can be leveraged to eliminate historical dynamic objects.

**Color and depth rendering** To obtain the final formulation of implicit map representation, we adopt a two-layer shallow MLP to predict the geometric and appearance information, respectively. The geometry decoder outputs the predicted SDF value $s$ and a feature vector $\mathbf{h}$ at the point $\mathbf{x}$. The appearance decoder outputs the predicted RGB value $c$. Similar to Co-SLAM (Wang et al., 2023), we joint encode the coordinate encoding $\gamma(\mathbf{x})$ and parametric encoding $V_\alpha$ as:

$$f_\beta\left(\gamma(\mathbf{x}), V_\alpha(\mathbf{x})\right) \mapsto (\mathbf{h}, s), \quad f_\phi\left(\gamma(\mathbf{x}), \mathbf{h}\right) \mapsto \mathbf{c}, \tag{1}$$

where $\{\alpha, \beta, \phi\}$ are the learnable parameters. Following the volume rendering method in NeRF (Mildenhall et al., 2021b), we accumulate the predicted values along the viewing ray $\mathbf{r}$ at the current estimation pose $\xi_i$ to render the color and depth value as:

$$\hat{C}(\mathbf{r}) = \frac{1}{\sum_{i=1}^M w_i} \sum_{i=1}^M w_i \mathbf{c}_i, \quad \hat{D}(\mathbf{r}) = \frac{1}{\sum_{i=1}^M w_i} \sum_{i=1}^M w_i z_i, \tag{2}$$

where $w_i$ is the computed weight along the ray, $\mathbf{c}_i$ and $z_i$ are the color and depth value of the sampling point $\mathbf{x}_i$. Since we do not directly predict voxel density $\sigma$ like NeRF, here we need to convert the SDF values $s_i$ into weights $w_i$. Thus, we employ a straightforward bell-shaped function (Azinović et al., 2022), formulated as the product of two sigmoid functions $\sigma(\cdot)$.

$$w_i = \sigma\left(\frac{s_i}{tr}\right)\sigma\left(-\frac{s_i}{tr}\right), \quad \hat{D}_{var}(\mathbf{r}) = \frac{1}{\sum_{i=1}^M w_i} \sum_{i=1}^M w_i(\hat{D} - z_i)^2, \tag{3}$$

where $tr$ denotes the truncation distance with TSDF prediction, $\hat{D}_{var}$ is the depth variance along this ray. When possessing GT depth values, we opt for uniform point sampling near the surface rather than employing importance sampling, with the aim of enhancing the efficiency of point sampling.

### 3.2 MOTION MASK GENERATION

For each input keyframe, we select its associated keyframes within a sliding window to compute the dense optical flow warping set $\mathcal{S}$. Note that optical flow estimation is conducted solely on keyframes, thereby optimizing system efficiency. To separate the ego-motion of dynamic objects, we additionally estimate the fundamental matrix $\boldsymbol{F}$ with inliers sampled from the matching set $\mathcal{S}$. Given any matching points $\boldsymbol{o}_{ji}, \boldsymbol{o}_{ki}$ within $\mathcal{S}$, we utilize matrix $\boldsymbol{F}$ to compute the Sampson distance between corresponding points and their epipolar lines. By setting a suitable threshold $e_{th}$, we derive the warp mask $\widehat{\mathcal{M}}_{j,k}^{wf}$ corresponding to dynamic objects as:

$$\widehat{\mathcal{M}}_{j,k}^{wf} : \left\{ \bigcap_{i=1}^M \mathbf{1}\left(\frac{\boldsymbol{o}_{ji}^T \boldsymbol{F} \boldsymbol{o}_{ki}}{\sqrt{A^2 + B^2}} < e_{th}\right) \otimes \boldsymbol{I}_{m \times n} \,\Big|\, \forall (\boldsymbol{o}_{ji}, \boldsymbol{o}_{ki}) \in \mathcal{S} \right\} \tag{4}$$

where $A, B$ denotes the coefficients of the epipolar line, and $m, n$ represents the size of the warp mask, aligning with the current frame image's dimensions. Additionally, $j$ and $k$ stand for the keyframe ID, illustrating the optical flow mask warping process from the k-th to the j-th keyframe. As illustrated in Fig. 1, to derive a more precise motion mask, we consider the spatial coherence of dynamic object motions within a sliding window of length $N$ and iteratively optimize the current

motion mask. Subsequently, we integrate the warp mask and segment mask to derive the final motion mask $\widehat{\mathcal{M}}_j$ as:

$$\widehat{\mathcal{M}}_j = \widehat{\mathcal{M}}_{j,k}^{wf} \otimes \widehat{\mathcal{M}}_{j,k-1}^{wf} \otimes \widehat{\mathcal{M}}_{j,k-2}^{wf} \cdots \otimes \widehat{\mathcal{M}}_{j,k-N}^{wf} \cup \widehat{\mathcal{M}}_j^{sg}, \tag{5}$$

where $\otimes$ represents the mask fusion operation, which is applied when pixels corresponding to a specific motion mask have been continuously observed for a duration exceeding a certain threshold $o_{th}$ within a sliding window. Note that we do not focus on the specific structure of the segment or optical flow network. Instead, we aim to introduce a general motion mask fusing method for application in NeRF-based SLAMs.

## 3.3 JOINT OPTIMIZATION

We introduce the details on optimizing the implicit scene representation and camera pose. Given a set of frames $\mathcal{F}$, we only predict the current camera pose represented with lie algebra $\xi_i$ in tracking process. Moreover, we utilize the global bundle adjustment (GBA) (Wang et al., 2021; Lin et al., 2021; Bian et al., 2023) to jointly optimize the sampled camera pose and the implicit mapping.

### 3.3.1 MAPPING

**Photometric rendering loss**   To jointly optimize the scene representation and camera pose, we render depth and color in independent view as Eq. 6 comparing with the proposed ground truth map:

$$\mathcal{L}_{rgb} = \frac{1}{M} \sum_{i=1}^{M} \left\| \left( \hat{C}(\mathbf{r}) - C(\mathbf{r}) \right) \cdot \widehat{\mathcal{M}}_i(\mathbf{r}) \right\|_2^2,$$

$$\mathcal{L}_{depth} = \frac{1}{N_d} \sum_{\mathbf{r} \in N_d} \left\| \left( \frac{\hat{D}(\mathbf{r}) - D(\mathbf{r})}{\sqrt{\hat{D}_{var}(\mathbf{r})}} \right) \cdot \widehat{\mathcal{M}}_i(\mathbf{r}) \right\|_2^2, \tag{6}$$

where $C(\mathbf{r})$ and $D(\mathbf{r})$ denote the ground truth color and depth map corresponding with the given pose,s respectively. $M$ represents the number of sampled pixels in the current image. Note that only rays with valid depth value $N_d$ are considered in $\mathcal{L}_{depth}$. In contrast to existing methods, we introduce the motion mask $\widehat{\mathcal{M}}_j$ to remove sampled pixels within the dynamic object region effectively. Moreover, to improve the robustness of pose estimation, we add the depth variance $\hat{D}_{var}$ to reduce the weight of depth outliers.

**Geometric constraints**   Following the practice (Azinović et al., 2022), assuming a batch of rays $M$ within valid motion mask regions are sampled, we directly leverage the free space loss with truncation $tr$ to restrict the SDF values $s(\mathbf{x_i})$ as:

$$\mathcal{L}_{fs} = \frac{1}{M} \sum_{i=1}^{M} \frac{1}{|\mathcal{R}_{fs}|} \sum_{i \in \mathcal{R}_{fs}} (s(\mathbf{x}_i) - tr)^2, \quad [u_i, v_i] \subseteq (\widehat{\mathcal{M}}_i = 1). \tag{7}$$

It is unreasonable to employ a fixed truncation value to optimize camera pose and SDF values in dynamic environments simultaneously. To reduce the artifacts in occluded areas and enhance the accuracy of reconstruction, we further divide the entire truncation region near the surface into middle and tail truncation regions inspired by ESLAM (Johari et al., 2023) as:

$$\mathcal{L}_{sdf} = \frac{1}{M} \sum_{i=1}^{M} \frac{1}{|\mathcal{R}_{tr}|} \sum_{i \in \mathcal{R}_{tr}} (s(\mathbf{x}_i) - (D[u_i, v_i] - T \cdot tr))^2, \quad [u_i, v_i] \subseteq (\widehat{\mathcal{M}}_i = 1), \tag{8}$$

where $T$ denotes the ratio of the entire truncation length occupied by the middle truncation. Note that we use the different weights to adjust the importance of middle and tail truncation in camera tracking and mapping process. The overall loss function is finally formulated as the following minimization,

$$\mathcal{P}^* = \arg\min_{\mathcal{P}} \lambda_1 \mathcal{L}_{rgb} + \lambda_2 \mathcal{L}_{depth} + \lambda_3 \mathcal{L}_{fs} + \lambda_4 \mathcal{L}_{sdf\text{-}m} + \lambda_5 \mathcal{L}_{sdf\text{-}t}, \tag{9}$$

where $\mathcal{P} = \{\theta, \phi, \alpha, \beta, \gamma, \xi_i\}$ is the list of parameters being optimized, including fields feature, decoders, and camera pose.

### 3.3.2 CAMERA TRACKING PROCESS

The construction of implicit maps within dynamic scenes often encounters substantial noise and frequently exhibits a lack of global consistency. Existing methods (Zhu et al., 2022; Wang et al., 2023; Johari et al., 2023; Li et al., 2023) rely solely on rendering loss for camera pose optimization, which makes the system vulnerable and prone to tracking failures. To solve this problem, we introduce edge warp loss to enhance geometry consistency in data association between adjacent frames.

**Edge reprojection loss**    For a 2D pixel $p$ in frame $i$, we first define the warp function as DIM-SLAM (Li et al., 2023) to reproject it onto frame $j$ as follows:

$$\boldsymbol{p}_{i \to j} = f_{warp}\left(\xi_{ji}, \boldsymbol{p}_i, D(\boldsymbol{p}_i)\right) = \boldsymbol{K}\boldsymbol{T}_{ji}\left(\boldsymbol{K}^{-1}D(\boldsymbol{p}_i)\boldsymbol{p}_i^{homo}\right), \tag{10}$$

where $\boldsymbol{K}$ and $\boldsymbol{T}_{ji}$ represent the intrinsic matrix and the transformation matrix between frame $i$ and frame $j$, respectively. $\boldsymbol{p}_i^{homo} = (u, v, 1)$ is the homogeneous coordinate of $\boldsymbol{p}_i$. Since the edge are detected once and do not change forwards, we can precompute the distance map (DT) (Felzenszwalb & Huttenlocher, 2012) to describe the projection error with the closest edge. For a edge set $\mathcal{E}_i$ in frame $i$, we define the edge loss $\mathcal{L}_{edge}$ as

$$\mathcal{L}_{edge} = \sum_{\boldsymbol{p}_i \in \mathcal{E}_i} \rho(\mathcal{D}_j(f_{warp}\left(\xi_{ji}, \boldsymbol{p}_i, D(\boldsymbol{p}_i)\right)) \cdot \widehat{\mathcal{M}}_j), \tag{11}$$

where $\mathcal{D}_j$ denotes the DT map in frame $j$, and the $\rho$ is a Huber weight function to reduce the influence of large residuals. Moreover, we drop a potential outlier if the projection distance error is greater than $\delta_e$. The pose optimization problem is finally formulated as the following minimization,

$$\xi_{ji}^* = \underset{\xi_{ji}}{\operatorname{argmin}} \ \lambda\mathcal{L}_{edge}, \quad \text{if } j \notin \mathcal{K} \tag{12}$$

To further improve the accuracy and stability of pose estimation, we employ distinct methods for tracking keyframes and non-keyframes in dynamic scenes. Keyframe pose estimation utilizes the edge loss to establish the initial pose, followed by optimization (Eq. 9). For non-keyframe pose estimation, we optimize the current frame's pose related to the nearest keyframe (Eq. 12).

## 4 EXPERIMENTS

**Datasets**    We evaluate our method on two real-world public datasets: *TUM RGB-D* dataset (Sturm et al., 2012) and *BONN RGB-D Dynamic* dataset (Palazzolo et al., 2019). Both datasets capture indoor scenes using a handheld camera and provide the ground-truth trajectory.

**Metrics**    For evaluating pose estimation, we adopt the RMSE and STD of Absolute Trajectory Error (ATE) (Sturm et al., 2012). The estimated trajectory is oriented to align with the ground truth trajectory using the unit quaternions algorithm (Horn, 1987) before evaluation. We also use three metrics which are widely used for scene reconstruction evaluation following (Zhu et al., 2022; Li et al., 2023): (i) *Accuracy* (cm), (ii) *Completion* (cm), (iii) *Completion Ratio* ($< 5$cm %). Since the BONN-RGBD only provided the ground truth point cloud, we randomly sampled the 200,000 points from both the ground truth point cloud and the reconstructed mesh surface to compute the metrics. We remove unobserved regions that are outside of any camera's viewing frustum and conduct extra mesh culling to remove the noisy points external to the target scene (Wang et al., 2023).

**Implementation details**    We adopt Co-SLAM Wang et al. (2023) as the baseline in our experiments and run our RoDyn-SLAM on an RTX 3090Ti GPU at 10 FPS (without optical flow mask) on the Tum datasets, which takes roughly 4GB of memory in total. Specific to implementation details, we sample $N_t = 1024$ rays and $N_p = 85$ points along each camera ray with 20 iterations for tracking and 2048 pixels from every 5 $th$ frames for global bundle adjustment. We set loss weight $\lambda_1 = 1.0$, $\lambda_2 = 0.1$, $\lambda_3 = 10$, $\lambda_4 = 2000$, $\lambda_1 = 500$ to train our model with Adam (Kingma & Ba, 2014) optimizer. Please refer to the appendix for more specific experiment settings.

### 4.1 EVALUATION OF GENERATING MOTION MASK

Fig. 2 shows the qualitative results of the generated motion mask. We evaluated our method on the `balloon` and `move_no_box2` sequence of the *BONN* dataset. In these sequences, in addition to

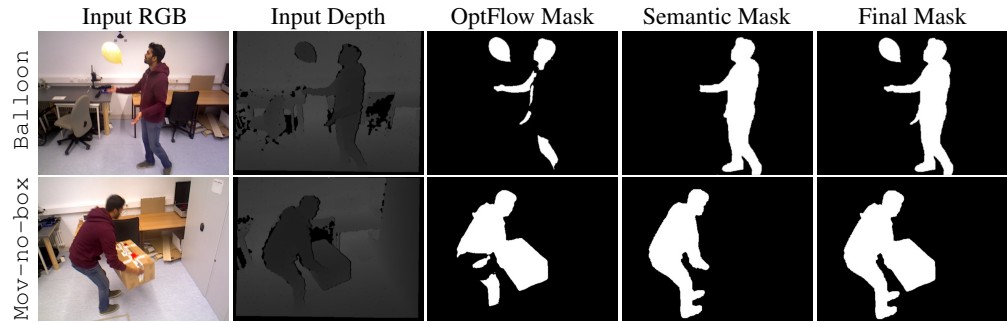

Figure 2: **Qualitative results of the generation motion mask.** By iteratively optimizing the optical flow mask, the fused optical mask can be more precise without noises. The semantic mask can only identify dynamic objects within predefined categories.

|  |  | ball | ball2 | ps_trk | ps_trk2 | mv_box2 | Avg. |
|---|---|---|---|---|---|---|---|
| iMAP* | **Acc.**[cm]↓ | 16.68 | 31.20 | 35.38 | 54.16 | 17.01 | 30.89 |
|  | **Comp.**[cm]↓ | 27.32 | 30.14 | 201.38 | 107.28 | 20.499 | 77.32 |
|  | **Comp. Ratio**[≤ 5cm%]↑ | 25.68 | 21.91 | 11.54 | 12.63 | 24.86 | 19.32 |
| NICE-SLAM | **Acc.**[cm]↓ | X | 24.30 | 43.11 | 74.92 | 17.56 | 39.97 |
|  | **Comp.**[cm]↓ | X | 16.65 | 117.95 | 172.20 | 18.19 | 81.25 |
|  | **Comp. Ratio**[≤ 5cm%]↑ | X | 29.68 | 15.89 | 13.96 | 32.18 | 22.93 |
| Vox-Fusion | **Acc.**[cm]↓ | 85.70 | 89.27 | 208.03 | 162.61 | 40.64 | 117.25 |
|  | **Comp.**[cm]↓ | 55.01 | 29.78 | 279.42 | 229.79 | 28.40 | 124.48 |
|  | **Comp. Ratio**[≤ 5cm%]↑ | 3.88 | 11.76 | 2.17 | 4.55 | 14.69 | 7.41 |
| Co-SLAM | **Acc.**[cm]↓ | 10.61 | 14.49 | 26.46 | 26.00 | 12.73 | 18.06 |
|  | **Comp.**[cm]↓ | 10.65 | 40.23 | 124.86 | 118.35 | 10.22 | 60.86 |
|  | **Comp. Ratio**[≤ 5cm%]↑ | 34.10 | 3.21 | 2.05 | 2.90 | 39.10 | 16.27 |
| ESLAM | **Acc.**[cm]↓ | 17.17 | 26.82 | 59.18 | 89.22 | 12.32 | 40.94 |
|  | **Comp.**[cm]↓ | 9.11 | 13.58 | 145.78 | 186.65 | 10.03 | 73.03 |
|  | **Comp. Ratio**[≤ 5cm%]↑ | 47.44 | **47.94** | 20.53 | 17.33 | 41.41 | 34.93 |
| Ours(RoDyn-SLAM) | **Acc.**[cm]↓ | **10.60** | **13.36** | **10.21** | **13.77** | **11.34** | **11.86** |
|  | **Comp.**[cm]↓ | **7.15** | **7.87** | **27.70** | **18.97** | **6.86** | **13.71** |
|  | **Comp. Ratio**[≤ 5cm%]↑ | **47.58** | 40.91 | **34.13** | **32.59** | **45.37** | **40.12** |

Table 1: **Quantitative results on several dynamic scene sequences in the *BONN-RGBD* dataset**. Reconstruction errors are reported with the subsampling GT point cloud using a laser scanner, which is provided in the original dataset. "X" denotes the tracking failures. The best results in RGB-D SLAMs are **bolded** respectively.

the movement of the person, there are also other dynamic objects associated with the person, such as balloons and boxes. As shown in Fig. 2 final mask part, our methods can significantly improve the accuracy of motion mask segmentation and effectively mitigate both false positives and false negatives issues in motion segmentation.

## 4.2 EVALUATION OF MAPPING AND TRACKING

**Mapping**    To better demonstrate the performance of our proposed system in dynamic scenes, we evaluate the mapping results from both qualitative and quantitative perspectives. Since the majority of dynamic scene datasets do not provide ground truth for static scene reconstruction, we adopt the *BONN* dataset to conduct quantitative analysis experiments. We compare our RoDyn-SLAM method against current state-of-the-art NeRF-based methods with RGB-D sensors, including NICE-SLAM (Zhu et al., 2022), iMap (Sucar et al., 2021), Vox-Fusion (Yang et al., 2022), ESLAM (Johari et al., 2023), and Co-SLAM (Wang et al., 2023), which are open source. The evaluation metrics have been mentioned above at the beginning of Section 4.

As shown in Tab. 1, our method outperforms most of the neural RGB-D slam systems on accuracy and completion. To improve the accuracy of pose estimation, we filter the invalid depth, which may reduce the accuracy metric on mapping evaluation. The visual comparison of reconstructed meshes with other methods (Curless & Levoy, 1996) is provided in Fig. 3. Our methods can generate a more accurate static mesh than other compared methods. Since the baseline methods (Wang et al., 2023) adopt the hash encoding to represent the implicit map, it may exacerbate the issue of the hash collisions in dynamic scenes and generate the hole in the reconstruction map.

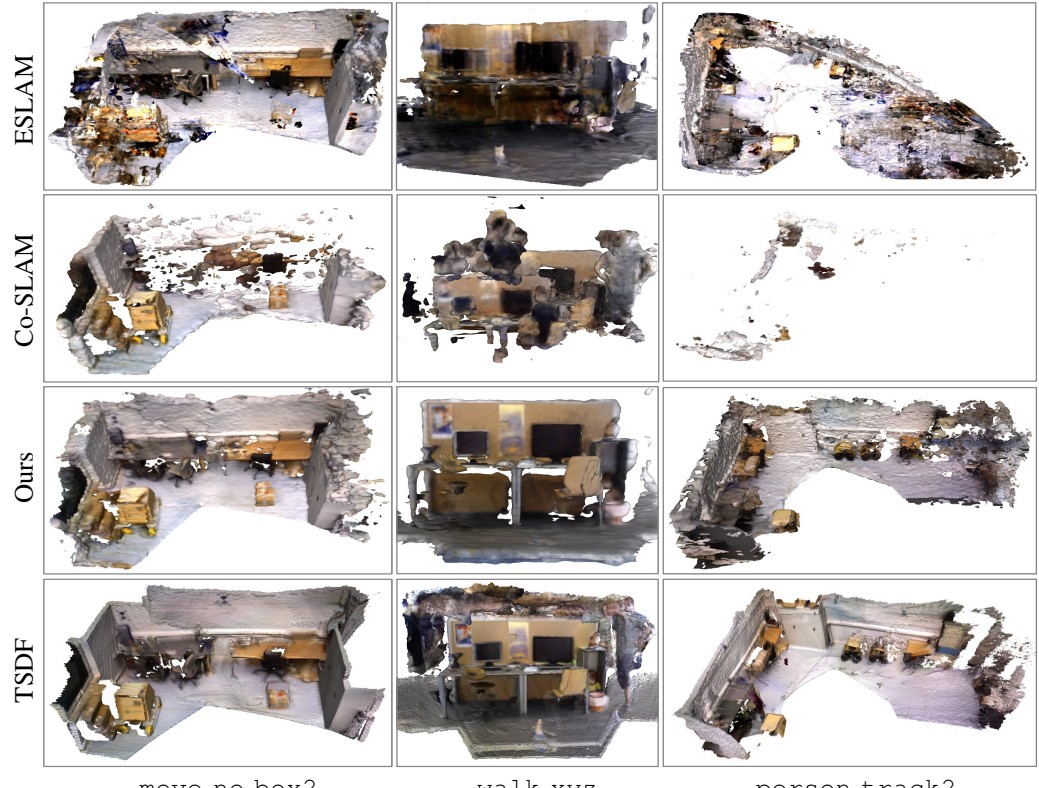

Figure 3: **Visual comparison of the reconstructed meshes on the *BONN* and *TUM RGB-D* datasets.** Our results are more complete and accurate without the dynamic object floaters. More examples are in the appendix.

| Method | f3/wk_xyz | | f3/wk_hf | | f3/wk_st | | f3/st_hf | | Avg. | |
|---|---|---|---|---|---|---|---|---|---|---|
| *Traditional SLAM methods* | *ATE* | *S.D.* | *ATE* | *S.D.* | *ATE* | *S.D.* | *ATE* | *S.D.* | *ATE* | *S.D.* |
| ORB-SLAM2(Mur-Artal & Tardós, 2017) | 45.9 | - | 35.1 | - | 9.0 | - | 2.0 | - | 23.0 | - |
| DVO-SLAM(Kerl et al., 2013) | 59.7 | - | 52.9 | - | 21.2 | - | 6.2 | - | 35.0 | - |
| DynaSLAM(Bescos et al., 2018) | **1.7** | - | 2.6 | - | **0.7** | - | 2.8 | - | 2.0 | - |
| ReFusion(Palazzolo et al., 2019) | 9.9 | - | 10.4 | - | 1.7 | - | 11.0 | - | 8.3 | - |
| MID-Fusion(Xu et al., 2019) | 6.8 | - | 3.8 | - | 2.3 | - | 3.1 | - | 4.0 | - |
| Droid-SLAM(Teed & Deng, 2021) | **1.7** | - | **2.0** | - | 1.2 | - | **2.0** | - | **1.7** | - |
| *NeRF based SLAM methods* | *ATE* | *S.D.* | *ATE* | *S.D.* | *ATE* | *S.D.* | *ATE* | *S.D.* | *ATE* | *S.D.* |
| iMAP*(Sucar et al., 2021) | 111.5 | 43.9 | X | X | 137.3 | 21.7 | 93.0 | 35.3 | 114.0 | 33.6 |
| NICE-SLAM(Zhu et al., 2022) | 113.8 | 42.9 | X | X | 88.2 | 27.8 | 45.0 | 14.4 | 82.3 | 28.4 |
| Vox-Fusion(Yang et al., 2022) | 146.6 | 32.1 | X | X | 109.9 | 25.5 | 89.1 | 28.5 | 115.2 | 28.7 |
| Co-SLAM(Wang et al., 2023) | 51.8 | 25.3 | 105.1 | 42.0 | 49.5 | 10.8 | 4.7 | 2.2 | 52.8 | 20.0 |
| ESLAM(Johari et al., 2023) | 45.7 | 28.5 | 60.8 | 27.9 | 93.6 | 20.7 | **3.6** | **1.6** | 50.9 | 19.7 |
| RoDyn-SLAM(Ours) | **8.3** | **5.5** | **5.6** | **2.8** | 1.7 | 0.9 | 4.4 | 2.2 | **5.0** | **2.8** |

Table 2: **Camera tracking results on several dynamic scene sequences in the *TUM RGB-D* dataset**. "∗" denotes the version reproduced by NICE-SLAM. "X" and "-" denote the tracking failures and absence of mention, respectively. The metric unit is [cm]. Please refer to the appendix for more results.

**Tracking** To evaluate the accuracy of camera tracking in dynamic scenes, we compare our methods with the recent neural RGB-D SLAM methods and traditional SLAM methods like ORB-SLAM2 (Mur-Artal & Tardós, 2017), DVO-SLAM (Kerl et al., 2013), Droid-SLAM (Teed & Deng, 2021) and traditional dynamic SLAM like DynaSLAM (Bescos et al., 2018), MID-Fusion (Xu et al., 2019), and ReFusion (Palazzolo et al., 2019).

As shown in Tab. 2, we report the results on three highly dynamic sequences and one slightly dynamic sequence from TUM RGB-D dataset. Our system achieves advanced tracking performance owing to the motion mask filter and edge-based optimization algorithm. We have also evaluated the tracking performance on the more complex and challenging BONN RGB-D dataset, as illustrated in Tab. 3. In more complex and challenging scenarios, our method has achieved superior results. While

| Method | balloon | | balloon2 | | ps_track | | ps_track2 | | Avg. | |
|---|---|---|---|---|---|---|---|---|---|---|
| *Traditional SLAM methods* | *ATE* | *S.D.* | *ATE* | *S.D.* | *ATE* | *S.D.* | *ATE* | *S.D.* | *ATE* | *S.D.* |
| DynaSLAM(Bescos et al., 2018) | **3.0** | - | **2.9** | - | 6.1 | - | **7.8** | - | **5.0** | - |
| ReFusion(Palazzolo et al., 2019) | 17.5 | - | 25.4 | - | 28.9 | - | 46.3 | - | 29.6 | - |
| Droid-SLAM(Teed & Deng, 2021) | 4.0 | - | 3.8 | - | **4.3** | - | 16.3 | - | 7.1 | - |
| *NeRF based SLAM methods* | *ATE* | *S.D.* | *ATE* | *S.D.* | *ATE* | *S.D.* | *ATE* | *S.D.* | *ATE* | *S.D.* |
| iMAP*(Sucar et al., 2021) | 14.9 | 5.4 | 67.0 | 19.2 | 28.3 | 12.9 | 52.8 | 20.9 | 40.7 | 14.6 |
| NICE-SLAM(Zhu et al., 2022) | X | X | 66.8 | 20.0 | 54.9 | 27.5 | 45.3 | 17.5 | 55.7 | 21.7 |
| Vox-Fusion(Yang et al., 2022) | 65.7 | 30.9 | 82.1 | 52.0 | 128.6 | 52.5 | 162.2 | 46.2 | 109.6 | 45.4 |
| Co-SLAM(Wang et al., 2023) | 28.8 | 9.6 | 20.6 | 8.1 | 61.0 | 22.2 | 59.1 | 24.0 | 42.4 | 16.0 |
| ESLAM(Johari et al., 2023) | 22.6 | 12.2 | 36.2 | 19.9 | 48.0 | 18.7 | 51.4 | 23.2 | 39.6 | 18.5 |
| RoDyn-SLAM(Ours) | **7.9** | **2.7** | **11.5** | **6.1** | **14.5** | **4.6** | **13.8** | **3.5** | **11.9** | **4.2** |

Table 3: **Camera tracking results on several dynamic scene sequences in the *BONN RGB-D* dataset**. "∗" denotes the version reproduced by NICE-SLAM. "X" and "-" denote the tracking failures and absence of mention, respectively. The metric unit is [cm]. Please refer to the appendix for more results.

there is still some gap compared to the more mature and robust traditional dynamic SLAM methods, our systems can drive the dense and textural reconstruction map to finish the more complex robotic navigation tasks.

## 4.3 ABLATION STUDY

To demonstrate the effectiveness of the proposed methods in our system, we perform the ablation studies on seven representative sequences of the *BONN* dataset. We compute the average ATE and STD results to show how different methods affect the overall system performance. The results presented in Tab. 4 demonstrate that all the proposed methods are effective in camera tracking. This suggests that fusing the optical flow mask and semantic motion mask can promote better pose estimation. At the same time, leveraging a divide-and-conquer pose optimization can effectively improve the robustness and accuracy of camera tracking.

| | ATE (m) ↓ | STD (m) ↓ |
|---|---|---|
| w/o Seg mask | 0.3089 | 0.1160 |
| w/o Flow mask | 0.1793 | 0.0739 |
| w/o Edge opt. | 0.2056 | 0.0829 |
| RoDyn-SLAM | **0.1354** | **0.0543** |

Table 4: **Ablation study of proposed methods**

| | Tracking (ms) | Mapping (ms) |
|---|---|---|
| NICE-SLAM | 3535.67 | 3055.58 |
| ESLAM | 1002.52 | 703.69 |
| Co-SLAM | 174.47 | **565.50** |
| RoDyn-SLAM | **159.06** | 675.08 |

Table 5: **Time comparision of different methods.**

## 4.4 TIME CONSUMPTION ANALYSIS

As shown in Tab. 5, we report time consumption (per frame) of the tracking and mapping without computing semantic segmentation and optical flow. All the results were obtained using an experimental configuration of sampled 1024 pixels and 20 iterations for tracking and 2048 pixels and 40 iterations for mapping, with an RTX 3090 GPU. Despite incorporating additional methods for handling dynamic objects, our system maintains a comparable level of computational cost to that of Co-SLAM.

## 5 CONCLUSION

We present RoDyn-SLAM, a novel dense RGB-D SLAM with neural implicit representation for dynamic environments. The proposed system is able to estimate camera poses and recover 3D geometry in this challenging setup thanks to the motion mask generation that successfully filters out dynamic regions. To further improve the stability and robustness of pose optimization, a divide-and-conquer pose optimization algorithm is designed to enhance the geometry consistency between keyframe and non-keyframe with the edge warp loss. The experiment results demonstrate that RoDyn-SLAM achieves state-of-the-art performance among recent neural RGB-D methods in both accuracy and robustness. In future work, a more robust keyframe management method is a promising direction to improve the system further.

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

## A APPENDIX

### A.1 ADDITIONAL IMPLEMENTATION DETAILS

**General configuration** Since the code base of our method is Co-SLAM (Wang et al., 2023), we use a similar configuration for scene representation. We utilize a multi-resolution hash grid with 16 levels, ranging from $R_{min} = 16$ to $R_{max}$, with a maximum voxel size of 2cm for determining $R_{max}$. The voxel size of the color encoder is 0.04 cm, and for the SDF encoder, it is 0.02 cm. We also employ the OneBlob encoding method with 16 bins. The above process is implemented leveraging the tiny-cuda-nn lib. The shallow MLP is designed with two hidden layers, each having a dimension of 32. The truncation distance $tr$ is set to 5cm.

**TUM RGB-D dataset** We use a learning rate of 1e-2 for the camera tracking process. To obtain more accurate joint optimization results, we employ the learning rate of 1e-3 in the mapping process. The weights of each loss are $\lambda_{rgb} = 1.0$, $\lambda_{depth} = 0.1$, $\lambda_{sdf\_m} = 2000$, $\lambda_{sdf\_t} = 500$, $\lambda_{fs} = 10$ . In the tracking process, we iterate the optimization process 20 times. In the edge extraction process, we utilize the Canny edge detection algorithm with a low threshold of 70 and a high threshold of 90.

**BONN RGB-D dataset** We also use a learning rate of 1e-2 and 1e-3 in the tracking and mapping process, respectively. The weights of each loss are $\lambda_{rgb} = 20.0$, $\lambda_{depth} = 0.01$, $\lambda_{sdf\_m} = 2000$, $\lambda_{sdf\_t}$ = 500, $\lambda_{fs} = 5$ . We also iterate the optimization process 20 times in the tracking process, like TUM dataset.

**Joint Optimization** To obtain a more consistent implicit map, we joint optimize the sampled camera pose and implicit map utilizing GBA operation. Before performing GBA, we first jointly optimize the camera pose and implicit map for the current frame. We resample the current keyframe with the $N_i = 2048$ sample points located out of the refined motion mask and perform optimization for 30 iterations, incorporating edge reprojection loss and rendering loss. In the GBA stage, we fixed the current frame pose and sample pixels as much as possible from the subset of historical keyframes. The number of sampled pixels in the current frame is attenuated based on the distance interval from the initial keyframe. When the system tracks more than 100 frames, the number of sampled points for the current frame will be reduced to 100, maintaining a minimum sampling point count. As we cannot perform loop closure operations like traditional SLAM, we do not introduce inter-frame constraints based on edge warp loss. Instead, we employ render loss for the GBA operation.

| Metric | Definition |
|--------|------------|
| Acc. | $\operatorname*{mean}_{\mathbf{p} \in P} \left( \min_{\mathbf{p}^* \in P^*} \|\mathbf{p} - \mathbf{p}^*\|_1 \right)$ |
| Comp. | $\operatorname*{mean}_{\mathbf{p}^* \in P^*} \left( \min_{\mathbf{p} \in P} \|\mathbf{p} - \mathbf{p}^*\|_1 \right)$ |
| Comp. Ratio[$\leq$ 5cm%] | $\operatorname*{mean}_{\mathbf{p}^* \in P^*} \left( \min_{\mathbf{p} \in P} \|\mathbf{p} - \mathbf{p}^*\|_1 < 0.05 \right)$ |
| ATE(RMSE) | $\frac{1}{N} \sum_{i=1}^{N} \sqrt{\frac{1}{K} \sum_{k=1}^{K} \left\| \mathbf{x}_{k,i} \boxminus \hat{\mathbf{x}}_{k,i}^{+} \right\|_2^2}$ |

Table 6: **Evaluation Metrics.** We show the evaluation metrics with their definitions that we use to measure reconstruction quality. $P$ and $P^*$ are the point clouds sampled from the predicted and the ground truth mesh.

### A.1.1 EXPERIMENTS METRICS DESCRIPTION

The equation for evaluating tracking and mapping performance is shown as Tab. 6, including *Accuracy*, *Completion*, *Completion Ratio*, and *Absolute Trajectory Error (ATE)*. *Accuracy* measures the proximity of reconstructed points to the ground truth and is defined as the mean distance between reconstructed points and the ground truth. *Completion* assesses the degree to which ground truth points are successfully recovered and is defined as the mean distance between ground truth points and the reconstructed points. *Completion Ratio* represents the percentage of the points located in the reconstructed mesh with the nearest points on the ground truth mesh within a 5 cm threshold. *ATE* represents the mean of absolute trajectory error between the estimated and ground truth poses under all measuring locations. In this paper, we evaluate our reconstructed mesh using the ground truth point cloud of the whole scene provided by the BONN Dataset. Specifically, we first downsample and filter the dense point cloud based on camera view projection to extract static point clouds for each single small scene. Then, we calculate the transformation matrix to align the reconstructed mesh with the ground truth mesh. Finally, we calculate the reconstruction metrics mentioned above by randomly sampling 200000 points from the point cloud and mesh surfaces. Note that this strategy may differ from the previous mesh-based sampling evaluation methods.

### A.1.2 SYSTEM

This section introduces several essential operations for constructing our comprehensive visual SLAM system.

**Initialization** When the first frame comes in, we initialize the implicit map using the ground truth pose to establish the correct scale. The multi-resolution hash encoding and MLP decoders will then be optimized over $N_i$ iterations, following Eq. 6. After the initialization, the first frame will be inserted into the keyframe set and held fixed. For dynamic objects that appear in the first frame, we directly utilize a semantic segmentation mask $\widehat{\mathcal{M}}_1^{sg}$ to filter the sampled rays. Thus, the potential influence of dynamic objects will be mitigated during the initial tracking period.

**KerFrame selection** Our system evenly inserts keyframes to ensure adequate observations and data associations between consecutive frames. We also try the other inserted way, such as based on the motion distance or the overlap between the current frame and the former keyframe. We find that these methods can not result in a significant improvement in accuracy. Therefore, we adopt the simple method to handle keyframe insertion in dynamic scenes. We also store a subset of pixels to represent each keyframe like (Wang et al., 2023). Compared with the Co-SLAM, we resample the pixels located out of the refined motion mask to represent the current keyframe in mapping process. For the pose estimation in dynamic scenes, storing only 5% representative points for each keyframe is insufficient, leading to the degeneracy of implicit representation and inadequate joint optimization. Therefore, we store the 50% of the pixels located in valid sampling regions to generate a subset that represents the keyframes. We also try to resample the high gradient point located on the

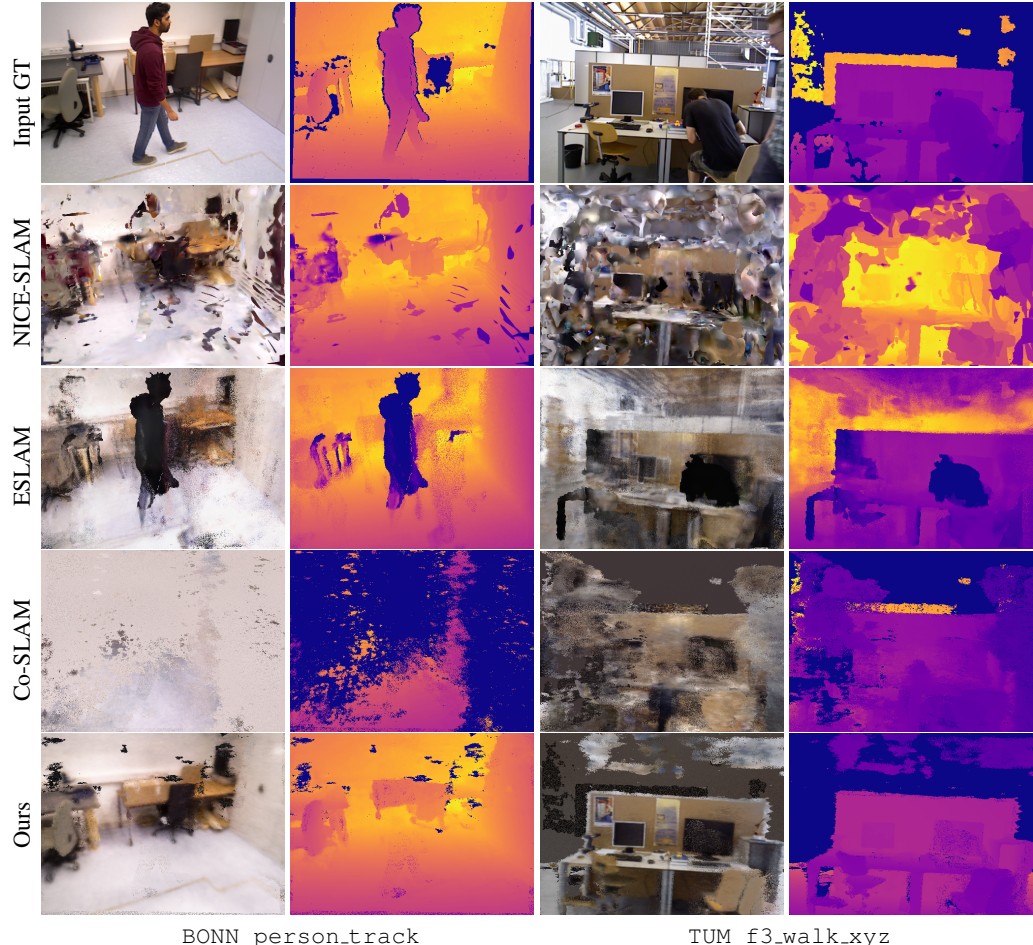

BONN person_track                    TUM f3_walk_xyz

Figure 4: **Visual comparison of the rendering image on the *TUM* and *BONN* datasets.**

edge to improve the representative capabilities. However, the results of pose estimation significantly degrade. Thus, we finally use uniform sampling to generate the subset of pixels to represent each keyframe.

**Depth edge filtering** Since the edges of dynamic objects are difficult to segment precisely, it causes significant depth errors along the object boundaries, which is destructive for pose optimization and static background reconstruction. For each generation of motion mask, we calculate the DT map $\mathcal{D}_j^{mask}$ and exclude sampled rays for which the distance to the mask edge is less than $\delta$. Furthermore, we exclude rays with no ground truth depths and outlier pixels from each optimization step. A pixel is classified as an outlier when the disparity between its measured depth and rendered depth exceeds six times the median rendered depth error within the batch.

A.2    ADDITIONAL EXPERIMENTAL RESULTS ON TRACKING AND MAPPING

A.2.1    RENDERING RESULTS

To further demonstrate the performance of static scene reconstruction, we compared the rendered image with the ground truth pose obtained from the generated static implicit map. We selected two challenging sequences, person_track from the BONN dataset and f3_walk_xyz from the TUM RGB-D dataset. These sequences involve complex motion, including dynamic objects (humans). As shown in Fig. 4, our method achieves a favorable rendering performance while enjoying the benefits of the proposed methods. Meanwhile, our methods can fill the hole which can not

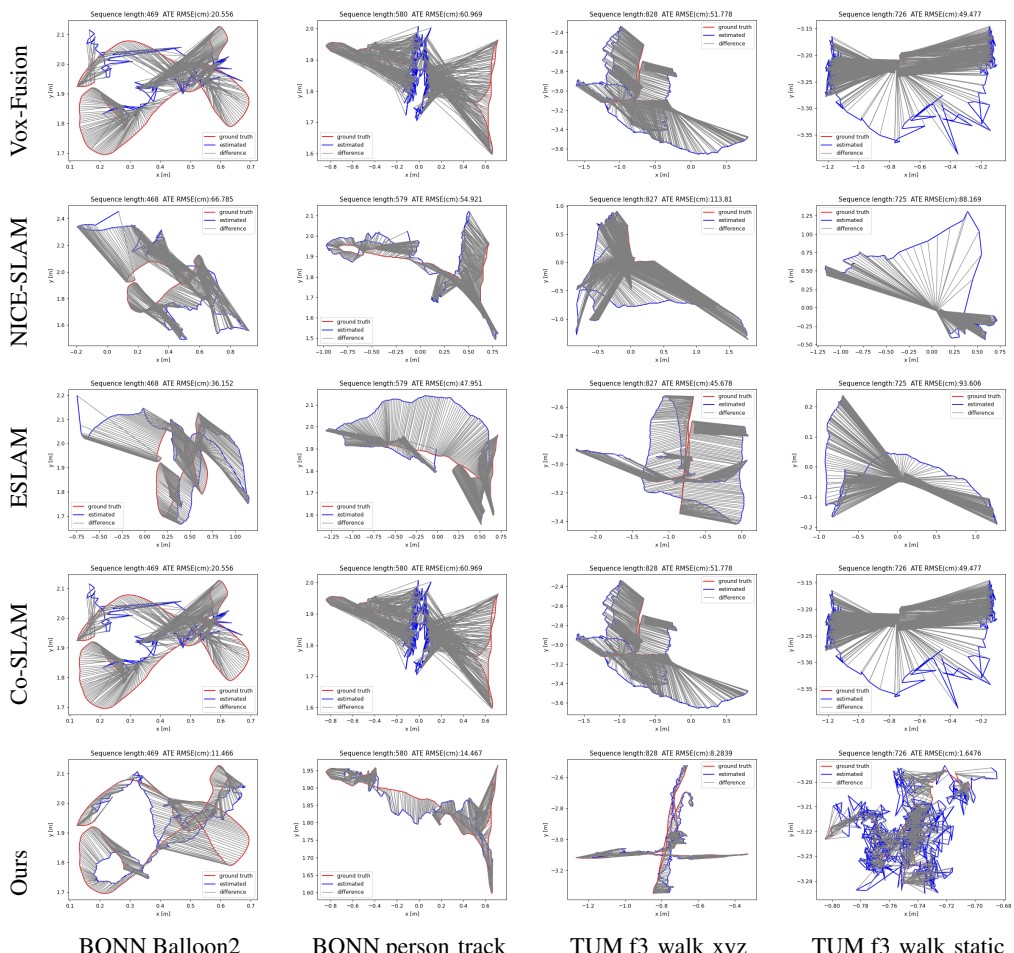

Figure 5: **Visual comparison of the trajectory error on the *TUM* and *BONN* RGB-D datasets.**

be captured in the original depth image. It can make the scene representation smoother and more complementing. We observed variations in rendering capabilities among different methods, which resulted in differences in the presentation quality. Note that our methods can be incrementally implemented in any existing baseline methods. Therefore, we don't focus on the actual performance of the code base Co-SLAM (Wang et al., 2023) but solely on the proposed methods's ability and effectiveness in addressing dynamic scene challenges.

### A.2.2 TRACKING RESULTS

Tab. 7 shows the external experiment results of camera tracking on the BONN and TUM RGB-D datasets. Compared with the recent neural RGB-D SLAM system, our methods achieve advanced performance during the tracking process under dynamic environments. `f2_desk_person` and `f3_long_office` represent the large scale lower dynamic scenes. Our methods can also predict the accurate camera pose without the strong dynamic assumption, which can further demonstrate the capability of pose estimation.

As shown in Fig. 5, we draw the absolute trajectory error map on the TUM and BONN RGB-D dataset, which has been mentioned in Experiments. The red line in the sub-figures represents the ground truth trajectory, the blue line represents the trajectory estimated by our system, and the errors between them are connected by gray lines. We can clearly observe that RoDyn-SLAM (ours) can achieve more robust and accurate pose estimation results under complete dynamic environments.

| Method | ball_trk | | ball_trk2 | | mv_box2 | | fr2/d_ps | | fr3/wk_rpy | | Avg. | |
|---|---|---|---|---|---|---|---|---|---|---|---|---|
| | *ATE* | *S.D.* | *ATE* | *S.D.* | *ATE* | *S.D.* | *ATE* | *S.D.* | *ATE* | *S.D.* | *ATE* | *S.D.* |
| iMAP * | 24.8 | 11.2 | 28.7 | 9.6 | 28.3 | 35.3 | 119.0 | 43.6 | 139.5 | 59.8 | 59.0 | 23.7 |
| NICE-SLAM | 21.2 | 13.1 | 27.4 | 9.7 | 31.9 | 13.6 | X | X | X | X | 26.8 | 12.1 |
| Vox-Fusion | 43.9 | 16.5 | 84.9 | 26.8 | 47.5 | 19.5 | X | X | X | X | 44.7 | 16.0 |
| Co-SLAM | 38.3 | 17.4 | 42.2 | 18.2 | 70.0 | 25.5 | 7.6 | 2.3 | 52.1 | 24.0 | 35.6 | 14.9 |
| ESLAM | **12.4** | 6.6 | 35.0 | **7.2** | 17.7 | 7.5 | X | X | 90.4 | 48.2 | 31.7 | 14.1 |
| Ours | 13.3 | **4.7** | **21.3** | 11.7 | **12.6** | **4.7** | **5.6** | **2.1** | **7.8** | **4.9** | **10.6** | **4.7** |

Table 7: **Camera tracking results on several dynamic scene sequences in the *BONN RGB-D* and *Tum* dataset**. "∗" denotes the version reproduced by NICE-SLAM. "X" denote the tracking failures. The best results are **bolded**. The metric unit is [cm].

## A.3 MORE EXPERIMENTS ON POSE OPTIMIZATION

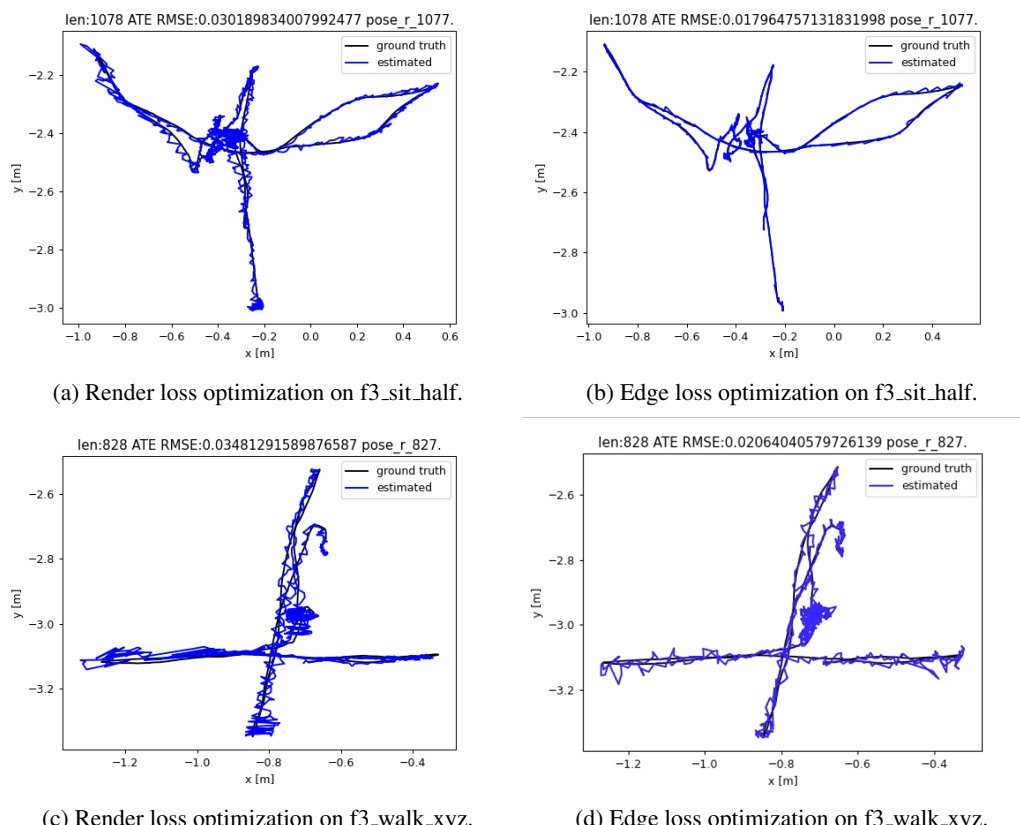

(a) Render loss optimization on f3_sit_half.

(b) Edge loss optimization on f3_sit_half.

(c) Render loss optimization on f3_walk_xyz.

(d) Edge loss optimization on f3_walk_xyz.

Figure 6: **Visual comparison of different pose optimization algorithm**

To further demonstrate the effectiveness of our proposed pose optimization algorithm, we evaluate the edge warp loss algorithm on non-keyframes with GT keyframe pose to compare the influence of two different optimization methods. We compute the RMSE of absolute trajectory error(ATE) with two different optimization algorithms. Note that the metric unit is [m] in this part. To better demonstrate the performance of our methods, we show the relative pose estimation results to evaluate the capability of pose estimation between adjacent frames. The final pose estimation results are shown as Fig. 6. Compared with rendering loss, our algorithm reduces the ATE (Absolute Trajectory Error) results in pose estimation by over 40 %, with an average improvement of 1.5 cm. We can observe that utilizing the edge warp loss significantly enhances the accuracy and robustness of pose estimation, resulting in a noticeable reduction in trajectory jitter. This also enhances the stability of pose estimation in dynamic scenes, thereby preventing pose estimation divergence during optimiza-

| | | ball | ball2 | ps_trk | ps_trk2 | mv_box2 | Avg. |
|---|---|---|---|---|---|---|---|
| iMAP* | Acc.[cm]↓ | 16.68 | 31.20 | 35.38 | 54.16 | 17.01 | 30.89 |
| | Comp.[cm]↓ | 27.32 | 30.14 | 201.38 | 107.28 | 20.499 | 77.32 |
| | Comp. Ratio[≤ 5cm%]↑ | 25.68 | 21.91 | 11.54 | 12.63 | 24.86 | 19.32 |
| NICE-SLAM | Acc.[cm]↓ | X | 24.30 | 43.11 | 74.92 | 17.56 | 39.97 |
| | Comp.[cm]↓ | X | 16.65 | 117.95 | 172.20 | 18.19 | 81.25 |
| | Comp. Ratio[≤ 5cm%]↑ | X | 29.68 | 15.89 | 13.96 | 32.18 | 22.93 |
| Vox-Fusion | Acc.[cm]↓ | 85.70 | 89.27 | 208.03 | 162.61 | 40.64 | 117.25 |
| | Comp.[cm]↓ | 55.01 | 29.78 | 279.42 | 229.79 | 28.40 | 124.48 |
| | Comp. Ratio[≤ 5cm%]↑ | 3.88 | 11.76 | 2.17 | 4.55 | 14.69 | 7.41 |
| Co-SLAM | Acc.[cm]↓ | 10.61 | 14.49 | 26.46 | 26.00 | 12.73 | 18.06 |
| | Comp.[cm]↓ | 10.65 | 40.23 | 124.86 | 118.35 | 10.22 | 60.86 |
| | Comp. Ratio[≤ 5cm%]↑ | 34.10 | 3.21 | 2.05 | 2.90 | 39.10 | 16.27 |
| ESLAM | Acc.[cm]↓ | 17.17 | 26.82 | 59.18 | 89.22 | 12.32 | 40.94 |
| | Comp.[cm]↓ | 9.11 | 13.58 | 145.78 | 186.65 | 10.03 | 73.03 |
| | Comp. Ratio[≤ 5cm%]↑ | 47.44 | 47.94 | 20.53 | 17.33 | 41.41 | 34.93 |
| TSDF-Fusion* (w/gt pose&mask) | Acc.[cm]↓ | 9.51 | 9.52 | 10.87 | 12.55 | 7.54 | 10.00 |
| | Comp.[cm]↓ | 11.46 | 9.96 | 35.72 | 33.79 | 8.41 | 19.87 |
| | Comp. Ratio[≤ 5cm%]↑ | 44.35 | 43.16 | 34.22 | 31.03 | 52.28 | 41.01 |
| Ours(RoDyn-SLAM) | Acc.[cm]↓ | 10.60 | 13.36 | 10.21 | 13.77 | 11.34 | 11.86 |
| | Comp.[cm]↓ | 7.15 | 7.87 | 27.70 | 18.97 | 6.86 | 13.71 |
| | Comp. Ratio[≤ 5cm%]↑ | 47.58 | 40.91 | 34.13 | 32.59 | 45.37 | 40.12 |

Table 8: **Quantitative results on several dynamic scene sequences in the *BONN-RGBD* dataset**. Reconstruction errors are reported with the subsampling GT point cloud using a laser scanner, which is provided in the original dataset. "X" denotes the tracking failures. The best results in RGB-D SLAMs are highlighted as first , second

| Method | f1/rpy | | f1/xyz | | f2/rpy | | f3/l_o | | Avg. | |
|---|---|---|---|---|---|---|---|---|---|---|
| *NeRF based SLAM methods* | *ATE* | *S.D.* | *ATE* | *S.D.* | *ATE* | *S.D.* | *ATE* | *S.D.* | *ATE* | *S.D.* |
| iMAP*(Sucar et al., 2021) | 16.0 | 13.8 | 7.9 | 7.3 | 2.4 | 1.3 | 7.1 | 3.6 | 8.4 | 6.5 |
| NICE-SLAM(Zhu et al., 2022) | 3.4 | 2.5 | 4.6 | 3.8 | 8.1 | 5.0 | X | X | 5.4 | 3.8 |
| Vox-Fusion(Yang et al., 2022) | 4.3 | 3.0 | 1.8 | 0.9 | 1.8 | 0.9 | 2.7 | 1.3 | 2.7 | 1.6 |
| Co-SLAM(Wang et al., 2023) | 3.9 | 2.8 | 2.3 | 1.2 | 2.5 | 1.4 | 3.3 | 1.9 | 3.0 | 1.9 |
| ESLAM(Johari et al., 2023) | 2.2 | 1.2 | 1.1 | 0.6 | 1.2 | 0.5 | 2.8 | 1.0 | 1.9 | 0.9 |
| RoDyn-SLAM(Ours) | 2.8 | 1.5 | 1.5 | 0.8 | 1.7 | 0.8 | 2.8 | 1.6 | 2.2 | 1.2 |

Table 9: **Camera tracking results on several static scene sequences in the *TUM RGB-D* dataset**. "∗" denotes the version reproduced by NICE-SLAM. "X" and "-" denote the tracking failures and absence of mention, respectively. The metric unit is [cm].

tion. The experiment results further demonstrate the effectiveness of our proposed methods in both accuracy and robustness.

## A.4 MORE COMPARISON RESULTS ON RECONSTRUCTION

To better compare the reconstruction quality of current neural rgb-d SLAM methods in dynamic scenes, we also evaluate the TSDF-Fusion reconstruction performance with gt pose and motion mask in Tab. 8. Owing to the potential pose estimation error, our method does not match the accuracy of TSDF-fusion in reconstruction precision. Thanks to the geometry estimation capabilities of implicit neural representations for unobserved scene parts, our SLAM system achieves superior completeness in the reconstructed mesh compared to the traditional TSDF-Fusion method. Simultaneously, due to pose estimation errors, there exists a certain deviation between the reconstructed mesh from our method and the ground truth mesh, thereby causing the comparable performance of the completion ratio.

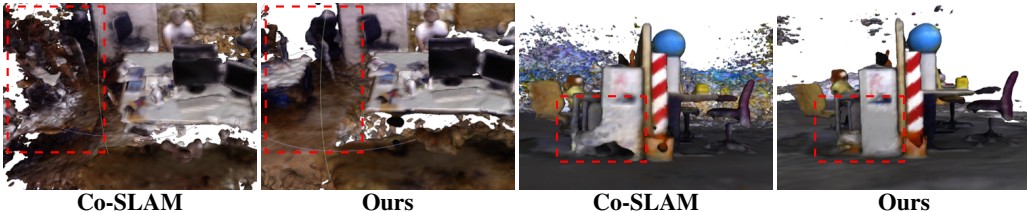

| Co-SLAM | Ours | Co-SLAM | Ours |

Figure 7: **Visual comparison of the detailed reconstructed meshes on several static scene in the *TUM RGB-D* dataset**. Compared with the baseline Co-SLAM Wang et al. (2023), our approach output more precise 3D structures in static scenes.(left: f1/rpy right: f3/long_office)

| Method | f3/wk_xyz | | f3/wk_hf | | f3/wk_st | | f3/st_hf | | Avg. | |
|---|---|---|---|---|---|---|---|---|---|---|
| *Traditional SLAM methods* | *ATE* | *S.D.* | *ATE* | *S.D.* | *ATE* | *S.D.* | *ATE* | *S.D.* | *ATE* | *S.D.* |
| ORB-SLAM2(Mur-Artal & Tardós, 2017) | 45.9 | - | 35.1 | - | 9.0 | - | 2.0 | - | 23.0 | - |
| DVO-SLAM(Kerl et al., 2013) | 59.7 | - | 52.9 | - | 21.2 | - | 6.2 | - | 35 | - |
| DVO-SLAM (w/mask)(Kerl et al., 2013) | 9.3 | - | 12.5 | - | 6.6 | - | 4.7 | - | 8.3 | - |
| ORB-SLAM3(Campos et al., 2021) | 28.1 | 12.2 | 30.5 | 9.0 | 2.1 | 1.1 | 2.6 | 1.6 | 15.9 | 6.0 |
| *NeRF based SLAM methods* | *ATE* | *S.D.* | *ATE* | *S.D.* | *ATE* | *S.D.* | *ATE* | *S.D.* | *ATE* | *S.D.* |
| iMAP*(Sucar et al., 2021) | 111.5 | 43.9 | X | X | 137.3 | 21.7 | 93.0 | 35.3 | 114.0 | 33.6 |
| NICE-SLAM(Zhu et al., 2022) | 113.8 | 42.9 | X | X | 88.2 | 27.8 | 45.0 | 14.4 | 82.3 | 28.4 |
| Vox-Fusion(Yang et al., 2022) | 146.6 | 32.1 | X | X | 109.9 | 25.5 | 89.1 | 28.5 | 115.2 | 28.7 |
| Co-SLAM(Wang et al., 2023) | 51.8 | 25.3 | 105.1 | 42.0 | 49.5 | 10.8 | 4.7 | 2.2 | 52.8 | 20.0 |
| ESLAM(Johari et al., 2023) | 45.7 | 28.5 | 60.8 | 27.9 | 93.6 | 20.7 | **3.6** | **1.6** | 50.9 | 19.7 |
| RoDyn-SLAM(Ours) | **8.3** | **5.5** | **5.6** | **2.8** | **1.7** | **0.9** | 4.4 | 2.2 | **5.0** | **2.8** |

Table 10: **Camera tracking results on several dynamic scene sequences in the *TUM RGB-D* dataset**. "*" denotes the version reproduced by NICE-SLAM. "X" and "-" denote the tracking failures and absence of mention, respectively. The metric unit is [cm].

## A.5 MORE EXPERIMENT RESULTS ON STATIC SCENES

To better illustrate the advantages of our proposed methods, we also evaluate the ATE performance on representative static sequences from the TUM dataset, as shown in Tab. 9. Actually, in static scenes, the motion mask generation strategy becomes irrelevant. Thus, it can sufficiently demonstrate the effectiveness of our proposed optimization strategy utilizing edge re-projection loss. Compared with our baseline methods Co-SLAM (Wang et al., 2023), our methods can effectively improve the accuracy of pose estimation. Notably, our proposed optimization algorithm is not restricted to a specific slam system. In most neural rgb-d slam systems, the performance of pose estimation heavily depends on implicit representation methods and the number of optimization iterations in the mapping and tracking process. Compared with the E-SLAM, our SLAM system requires less storage space and achieves real-time running at 10fps, despite several hours of training time needed for E-SLAM. Consequently, our method strikes a well-balanced compromise between accuracy and speed, demonstrating competitive performance of pose estimation when compared to state-of-the-art methods.

Since the TUM RGB-D dataset does not provide ground truth point clouds for evaluating the reconstruction quality, we visualize the reconstructed mesh and compare it with our baseline method, Co-SLAM, to showcase the effectiveness of our proposed methods. As shown in Fig. 7, our methods utilize a divide-and-conquer pose optimization algorithm, enabling more accurate pose estimation results and resulting in more precise and fewer floaters in the reconstructed mesh.

## A.6 MORE COMPARISON RESULTS WITH THE TRADITIONAL SLAM METHOD

As shown in Tab. 10, we also evaluate the pose estimation results of the representative traditional SLAM methods like ORB-SLAM3 and DVO-SLAM on the TUM dynamic datasets. Thanks to the introduced motion mask generation methods and the divide-and-conquer pose optimization algorithm, we can effectively improve the robustness and accuracy of pose estimation for current neural RGB-D SLAM system in dynamic scenes.

## A.7 MORE VISUALIZATION RESULTS

As shown in Fig. 8, we also visualize more reconstruction mesh results on the TUM and BONN RGB-D datasets. It is clear that our methods can significantly filter out the dynamic objects and recover more accurate and plentiful geometry structure details of static scenes.

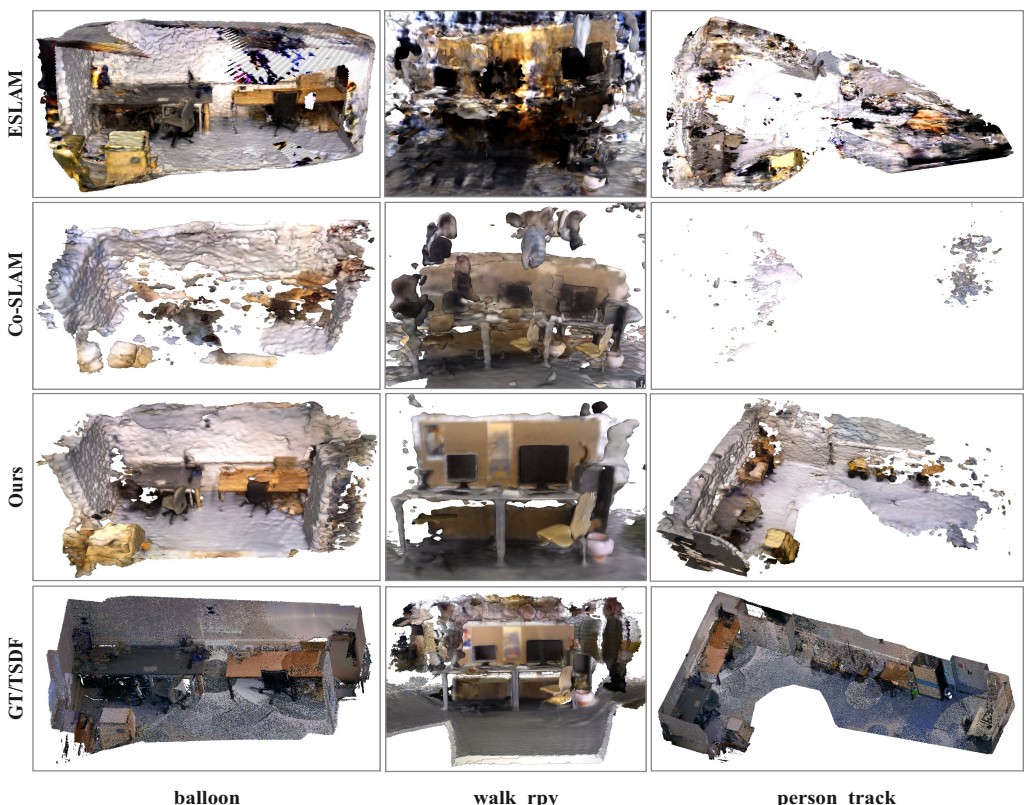

Figure 8: **Visual comparison of the reconstructed meshes on the *BONN* and *TUM RGB-D* datasets.** Our results are more complete and accurate without the dynamic object floaters.

