# OpenReview forum: "RoDyn-SLAM: Robust Dynamic Dense RGB-D SLAM with Neural Radiance Fields"
_ICLR.cc/2024/Conference — Submitted to ICLR 2024_

### Official Review · Reviewer_x3wF · 2023-10-24

**Soundness:** 3 good
**Presentation:** 3 good
**Contribution:** 3 good
**Rating:** 6
**Confidence:** 4

**Summary:**

Proposed a NeRF-based SLAM system (built upon Co-SLAM) to reconstruct the static 3D scene map in dynamic environments. To handle invalid sampling rays within dynamic objects, we filtered them out using a motion mask generation approach based on checking inliers and outliers using the fundamental matrix. Additionally, we introduced an Edge reprojection loss to remove the velocity constant assumption.

**Strengths:**

The paper is well-motivated and clearly written. It proposes techniques to address invalid sampling rays within dynamic objects, improving pose accuracy and robustness. Extensive evaluations support these claims, and the paper has a sufficient number of references.

**Weaknesses:**

Although I believe the paper is well-motivated and presents promising results in terms of both pose estimation and reconstruction, I have several concerns.
1. Doesn't introduce substantial architectural changes or novelty for NeRF-based SLAM. Such as multi-resolution hash encoding from instant ngp, joint optimization and store a subset of pixels to represent each keyframe from Co-SLAM, etc.
2. Enhancing additional losses (e.g., Edge reprojection loss) and masks to improve accuracy in dynamic object scenarios. How does the system's performance compare in terms of pose estimation and reconstruction in static scenes? Is it competitive with other SLAM systems?
3. I assume the GBA here is not an actual GBA like deployed in loop closure.

**Questions:**

The questions are listed in the weakness part.

---

> ### Author Response · Authors · 2023-11-22
> **Responce to reviewer x3wF**
>
> Thanks for the positive and detailed review as well as the valuable suggestions for improvement. We would like to address the reviewer's concerns as follows:
>
> **W1: More substantial architectural changes details**.
> Thanks for the suggestion. We have added the more substantial architectural details for our SLAM system in Appendix A.1 in the revised paper.
>
> **W2: More experiments in static scenes**.
> Thanks for the suggestion. To better demonstrate the system performance of pose estimation and reconstruction in a static scene, we choose the real-world RGB-D dataset [1] rather than the synthetic datasets [2,3]. Compared with the current the state of the art neural rgb-d SLAM, we evaluate the pose estimation and reconstruction results on three represented sequences. Additional detailed comparison results are available in Table 9 in Appendix A.5.
>
> Compared with our baseline methods Co-SLAM [4], `our method does not compromise the performance of the original SLAM methods in terms of tracking and mapping in static scenes`. In fact, it achieves competitive results.Notably, our proposed optimization algorithm is not restricted to a specific slam system. Thus, it can also be applied to other neural rgb-d slam methods to improve the data association between the inter-frame.
>
> Compared with the E-SLAM [5], our SLAM system requires less storage space and achieves real-time running at 10fps, despite several hours of training time needed for E-SLAM. Consequently, our method strikes a well-balanced compromise between accuracy and speed, demonstrating competitive performance of pose estimation when compared to state-of-the-art methods.
>
>
> > [1] J ̈urgen Sturm, Nikolas Engelhard, Felix Endres, Wolfram Burgard, and Daniel Cremers. A benchmark for the evaluation of rgb-d slam systems. In IROS, 2012.
>
> > [2] Julian Straub, Thomas Whelan, Lingni Ma, Yufan Chen, Erik Wijmans, Simon Green, Jakob J Engel, Raul Mur-Artal, Carl Ren, Shobhit Verma, et al. The replica dataset: A digital replica of indoor spaces. arXiv preprint, 2019.
>
> > [3] Dejan Azinovi ́c, Ricardo Martin-Brualla, Dan B Goldman, Matthias Nießner, and Justus Thies. Neural rgb-d surface reconstruction. In CVPR, 2022.
>
> > [4] Hengyi Wang, Jingwen Wang, and Lourdes Agapito. Co-slam: Joint coordinate and sparse parametric encodings for neural real-time slam. In CVPR, 2023.
>
> > [5] JMohammad Mahdi Johari, Camilla Carta, and Franc ̧ois Fleuret. Eslam: Efficient dense slam system based on hybrid representation of signed distance fields. In CVPR, 2023.
>
>
> **W3: GBA defination**.
> Yes, your assumption is correct. We do not contribute to the loop closure methods with neural radiance fields. In this paper, GBA represents an operation that samples rays from entire keyframes management set and performs a global bundle adjustment to reduce the pose estimation errors and obtain a more consistent implicit map representation. We are also considering incorporating a loop closure module in future work to implement GBA in loop closure and reduce accumulated errors.

---

### Official Review · Reviewer_az7R · 2023-10-29

**Soundness:** 3 good
**Presentation:** 3 good
**Contribution:** 3 good
**Rating:** 5
**Confidence:** 4

**Summary:**

The paper presents a novel RGB-D dynamic SLAM system featuring a neural radiance representation. The system builds upon the open-sourced RGB-D neural SLAM Co-SLAM, introducing a motion mask (comprising optical flow and semantic information) to filter out invalid dynamic rays during training.

**Strengths:**

1. The exploration of dynamic SLAM with neural radiance field representation is a relatively new and promising avenue.
2. The paper is well-written and easy to follow.
3. The evaluation results are visually compelling and present a convincing case for the proposed method.

**Weaknesses:**

A key concern in this paper is the insufficiently explained rationale for incorporating neural radiance field representation in dynamic SLAM, along with a noticeable absence of robust baseline comparisons during the evaluation.

Please see questions for details.

**Questions:**

1. While the adoption of neural radiance fields in dynamic SLAM is a novel exploration, the motivation behind this choice is not entirely clear. A more detailed explanation of the potential benefits compared to existing dense dynamic SLAM methods would enhance the paper's clarity.
2. The evaluation primarily compares the proposed method against static neural SLAM, which may not be entirely fair. It would be beneficial to include related works as stronger baselines, including traditional SLAM methods like MID-Fusion[1] and Droid-SLAM[2], as well as NeRF SLAM methods like vMAP[3] and BundleSDF[4].
3. Although Fig. 3 displays TSDF fusion results, these results are not reported in tables. Comparing the results with and without the motion mask, as proposed in this paper, using TSDF fusion would be valuable and should be considered a baseline method.
4. The paper's visualization results are impressive; however, for a SLAM system, it would be preferable to include videos or real-world demonstrations as additional supplementary material.

[1]Mid-fusion: Octree-based object-level multi-instance dynamic slam. ICRA 2019.
[2]Droid-slam: Deep visual slam for monocular, stereo, and rgb-d cameras. NeurIPS 2021.
[3]vMAP: Vectorised object mapping for neural field slam. CVPR 2023
[4]BundleSDF: Neural 6-DoF Tracking and 3D Reconstruction of Unknown Objects. CVPR 2023

---

> ### Author Response · Authors · 2023-11-22
> **Responce to reviewer az7R (1/2)**
>
> Thanks for the insightful and detailed review as well as the valuable suggestions for improvement. We would like to address the reviewer's concerns as follows:
>
> **W1: Motivation of neural RGB-D SLAM over dense RGB-D SLAM in dynamic scene**.
> Thanks for the suggestions. We have revised the introduction to motivate the benefits compared to existing dynamic dense RGB-D SLAM methods. All the modified text is highlighted in blue in the revised version.
>
> Compared with the existing dense RGB-D dynamic SLAM methods, the key advantage of neural RGB-D SLAM lies in its neural implicit representation method. The neural scene representations have attractive properties for mapping in dynamic scenes, including `improving noise and outlier handling, geometry estimation capabilities for unobserved scene parts, high-fidelity reconstructions with reduced memory usage, and the ability to generate high-quality static background images from novel views`. Moreover, neural radiance fields also open up possibilities for the subsequent development of dynamic SLAM, such as distilling perceptive information and accomplishing object-based tracking or reconstruction.
>
> Finally, please allow us to briefly introduce our motivation and contribution to building Rodyn-SLAM system. Existing neural RGB-D SLAM methods rely on a static environment assumption and do not work robustly within a dynamic environment due to the inconsistent observation of geometry and photometry. To tackle this problem, we propose a dynamic object identification method and filter the invalid sample ray to recover the static scene map. To further improve the accuracy of pose estimation in dynamic scenes, we propose novel and robust pose optimization methods utilizing edge projection loss to enhance data association between convisible frames.
>
>
> **W2: More experiment results with stronger baselines**.
> Thanks for the suggestion. Due to the current limitation of NeRF-based SLAM methods in handling dynamic scenes, we propose a general robust dynamic Nerf-based SLAM method RoDyn-SLAM, which constitutes the innovation of this paper. We have selected state-of-the-art NeRF-based SLAM methods as our benchmark for comparison, serving as robust baselines. Following your suggestions, we have added the results of MID-Fusion [1] and Droid-SLAM [2] in Table 2 and Table 3 in the revised paper, respectively. Due to the advanced factor graph optimization algorithm and extensive training data, Droid-SLAM has achieved superior pose estimation results in dynamic scenes compared to Nerf-based SLAM methods.
>
> `Bundle-SDF [3] and vMAP [4]` are object-based neural SLAM methods rather than a general nerf-based SLAM method. It relies on external conditions, such as the `availability of pose prior` or the requirement for the `camera to remain stationary`. Thus, we do not add the comparison results of these methods in the revised paper. The more detailed analysis is summarized as:
>
> * Bundle-SDF is designed for estimating the object pose and reconstructing the object mesh. Our approach aims to estimate the camera pose in dynamic scenes by leveraging a static background. The objectives of these two methods are distinct. Additionally, the validation datasets used for Bundle-SDF require the camera to remain stationary, allowing for the pose estimation of moving objects. The datasets we employed for evaluation do not meet these conditions.
>
> * vMAP primarily emphasizes joint reconstruction of objects, relying on estimated poses from other methods i.e. ORB-SLAM3 to provide the initial pose. Therefore, we believe that the direct comparison of pose optimazation is unfair. We attempted to make a comparison of reconstruction performance with vMAP given GT pose on TUM RGB-D dynamic sequence. However, the final reconstruction results for dynamic scenes were notably blurred. Actually, due to the absence of joint optimization between the camera pose and implicit map, the dynamic objects can not be filtered, resulting in significant blurriness and floaters in the reconstructed scene.
>
> > [1] Binbin Xu, Wenbin Li, Dimos Tzoumanikas, Michael Bloesch, Andrew Davison, and Stefan Leutenegger. Mid-fusion: Octree-based object-level multi-instance dynamic slam. In ICRA, 2019.
>
> > [2] Zachary Teed and Jia Deng. DROID-SLAM: Deep Visual SLAM for Monocular, Stereo, and RGB-D Cameras. In NeurIPS, 2021.
>
> > [3] DBowen Wen, Jonathan Tremblay, Valts Blukis, Stephen Tyree, Thomas M ̈uller, Alex Evans, Dieter Fox, Jan Kautz, and Stan Birchfield. Bundlesdf: Neural 6-dof tracking and 3d reconstruction of unknown objects. In CVPR, 2023.
>
> > [4] Xin Kong, Shikun Liu, Marwan Taher, and Andrew J Davison. vmap: Vectorised object mapping for neural field slam. In CVPR, 2023.

---

> > ### Author Response · Authors · 2023-11-22
> > **Responce to reviewer az7R (2/2)**
> >
> > **W3: TSDF fusion as a baseline**.
> >
> >
> > Thanks for the suggestion. `TSDF-fusion` is a traditional reconstruction method with `ground truth pose` as input rather than a SLAM method. We have reported the comparison results and analysis of TSDF-fusion in the revised version, available in Table 8 in Appendix A.4, aiming to address your concern. However, directly using this as a baseline method to compare with the other neural RGB-D slam method in dynamic scenes is not entirely fair. Because the `current mapping results` are built `based on an estimated pose rather than the ground truth pose`. The quality of mapping will be significantly affected by the error of pose estimation in the dynamic scene. This differs slightly from the quality evaluation of reconstruction in static environments.
> >
> > Finally, please allow us to explain why the comparison results for TSDF-Fusion are not included in the tables. The TUM dataset does not provide the ground truth mesh or point cloud for evaluating the reconstruction quality, and the BONN dataset only offers the point cloud captured by lidar scan. Thus, we evaluate the quantitative results in the BONN dataset. To better compare and visualize the reconstruction results, we utilize TSDF-fusion to generate the scene mesh as the ground truth mesh, leveraging the known ground truth pose.
> >
> > **W4: More visualization results on the real-world dataset**.
> > Thanks for the suggestion. Following your suggestions, we provide an additional video demo to visualize the pose estimation and reconstruction results. Please refer to the supplementary material. Notably, to better illustrate the effort of our proposed method, we have evaluated the performance of the SLAM system on real-world datasets such as TUM and BONN rgb-d datasets rather than the synthetic datasets in the original paper. These two datasets cover representative dynamic scenes in daily life. Thus, we believe it can demonstrate the real-world performance of our SLAM system.

---

> > ### Comment · Reviewer_az7R · 2023-11-22
> > **Revisions**
> >
> > Thanks a lot for the author's response.
> > I didn't see the revision. Could you please check if it has been uploaded?

---

> > > ### Author Response · Authors · 2023-11-22
> > >
> > > Thank you for your reminder. Upon reviewing the revision history, we confirm that the revision paper and video were uploaded at 7 pm. We have re-examined the submitted paper and supplementary materials. The uploaded PDF is the latest version, with the modified text highlighted in blue. The corresponding open access link is https://openreview.net/pdf?id=mmCIov21zD. If you have any other questions, please feel free to contact us.

---

### Official Review · Reviewer_H57b · 2023-11-01

**Soundness:** 3 good
**Presentation:** 4 excellent
**Contribution:** 3 good
**Rating:** 8
**Confidence:** 3

**Summary:**

This paper presents a implicit representation based RGB-D SLAM system, that is able to handle dynamic in the scene well. Two major features, namely 1) motion masking from both semantic and motion segmentation, and 2) per-frame tracking with edge consistency loss, are described in detail. Evaluation and ablation are sufficient, and the numbers seem pretty strong.

**Strengths:**

- This paper revisits an old topic in dynamic SLAM, i.e., getting rid of dynamic pixels before explicit / implicit optimization process. The idea of fusing motion segmentation mask over multiple keyframes is sound, and the evaluation reported are comprehensive and solid.
- As a system paper, the author did a great job covering both the overall system design, and the key components (masking and tracking) that contributes to the better performance of overall system. Writing and visualization are very clear to follow.

**Weaknesses:**

- While the overall writing is good, there are a few places that worthy of fix: e.g., in section 3.2 eq (4) there is no introduction on $j$ and $k$; also typos such as Camear in Fig 1. A thorough proofread is recommended.

**Questions:**

It's more of a general question on the subject of RGB-D only SLAM: modern visual-inertial solution has been proven to be very accurate and robust in providing highly accuracy 6DoF camera pose at a local scene. Therefore it sounds reasonable to formulate (implicit) mapping as a separate problem from pose tracking. How do you see this project going next on the mapping side?

---

> ### Author Response · Authors · 2023-11-22
> **Responce to reviewer H57b**
>
> Thanks for the positive and detailed review as well as the valuable suggestions for improvement. We would like to address the reviewer's concerns as follows:
>
> **W1: Lack introduction in Section 3.2 eq (4)**.
> Thank you for pointing it out. We have added the introduction of the meaning of variables $j$ and $k$ in Equation 4 to address confusion about the expression‘s significance. $j$ and $k$ stand for the keyframe ID, illustrating the optical flow mask warping process from the k-th to the j-th keyframe. All the modified text is highlighted in blue in the revised version.
>
> **W2: Typos in Fig 1**.
> Thank you for pointing out the spelling mistake in Fig 1. It is a typo and has been corrected. We have reviewed this paper to reduce occurrences of spelling issues in the revised paper.
>
> **Q1: The project going next on the (implicit) mapping**.
> It's my honor to discuss this question with you. We appreciate your opinion, and this is a question we have been contemplating recently as well. Actually, for the dynamic SLAM, it is crucial to `identify the dynamic objects` and `enhance the data association` in the tracking or mapping process. We follow this principle in designing the Rodyn-SLAM system, providing a specific technical solution based on neural RGB-D SLAM. While the method is relatively universal, minor adjustments (not based on depth information) may be necessary for different sensor inputs.
>
> In a sense, the coupled inertial measurement is also an observation to enhance the data association utilizing IMU preintegration theory in optimization process. However, it is insufficient to rely only on inter-frame constraints from IMU if the visual tracking part is not adjusted. Although the modern visual-inertial solution can improve the robustness and accuracy compared with the pure visual solution in dynamic environments, the precision may not satisfy the purpose of separate mapping.
>
> Formulate (implicit) mapping as a separate problem from pose tracking has a latent problem in which the tracking pose may not be consistent with the current mapping results. Certainly, without considering the above practical problems, we can also conduct research on the separate mapping in dynamic scenes. Since the background motion has been provided, independently estimating the pose and reconstructing the geometry structure of dynamic objects becomes a crucial and outstanding problem. It seems that employing `object-based neural implicit mapping and object-level tracking` could offer improved solutions to the aforementioned issues.

---

### Official Review · Reviewer_HXpV · 2023-11-22

**Soundness:** 3 good
**Presentation:** 3 good
**Contribution:** 2 fair
**Rating:** 5
**Confidence:** 5

**Summary:**

The paper presents a method for dealing with dynamic environments for the purpose of rib d slam.  A motion mask appears to be created from the optical flow field and a semantic mask.  A neural radiance field is computed and sampled rays that are invalid are used to help generate the motion mask.

**Strengths:**

The paper presents the algorithm used with a clear description.  Results are presented comparing to other techniques in the field.

**Weaknesses:**

Only 2 datasets are used for testing.
Some acronyms are not provided what is ATE? in the results section.
It does not appear that the authors address the degenerate cases for computing the fundamental matrix, how does their method handle this?
With the comparisons, you should have compared to orb slam 3 and possibly DVO slam, an indirect and direct traditional method.
There is no mention of computation times, can this run in real time and what type of computing resources are required for such.  With the computation of a radiance field and the extra processes, it is hard to see this functioning in real time

**Questions:**

see weaknesses

---

> ### Author Response · Authors · 2023-11-23
> **initial response to Reviewer HXpV**
>
> Thanks. As the Reviewer HXpV submitted the review comments very late (on the last day of author-reviewer discussion period), please allow some time for a more comprehensive reply.

---

> ### Author Response · Authors · 2023-11-23
> **Responce to reviewer HXpV (1/2)**
>
> We thank the reviewer for the valuable suggestions and address the reviewer’s concerns as follows:
>
> **Q1: Testing dataset**.
> Following traditional dynamic RGB-D SLAM methods [1,2,3], we assess the pose estimation results and reconstruction quality on the TUM RGB-D and BONN RGB-D datasets. As indicated in these papers, these datasets are sufficient to evaluate the performance of dynamic SLAM system. Additional evaluation results are presented in Appendix A.2, specifically in Tables 7 and 9 of the revised paper.
>
> **Q2: ATE meaning**.
> We have illustrated the meaning of ATE in the original paper. Please refer to the "Metric" part in the Experiment section for more details.
>
> **Q3: Comparison with ORB-SLAM3 and DVO SLAM**.
> As our method is a neural-based dynamic SLAM method, we selected the traditional dynamic SLAM methods, such as ReFusion [1], DynaSLAM [2], and MID-Fusion [3], as the baseline methods instead of the general static SLAM methods. ORB-SLAM3 [4] and DVO SLAM [5] rely on a static environment assumption and do not work robustly within a dynamic environment due to the inconsistent observation of geometry.
>
> DynaSLAM has showcased competitive pose estimation results in comparison to DVO-SLAM. We add the comparison results of DVO-SLAM in Table 2 in the revised paper, cited from DynaSLAM. Please refer to the original DynaSLAM paper for more details.
>
> ORB-SLAM3 primarily introduces multiple map fusion and tightly integrated visual-inertial fusion techniques via MAP estimation. However, it still depends on the assumption of a static environment, making it similar to ORB-SLAM2 [6] in this regard. We have included the ORB-SLAM2 pose estimation results in Table 2, cited from ReFusion. To address your concerns, we also evaluated the results of ORB-SLAM3 and included them in Table 10 in Appendix A.6.
>
> Our comparison results with ORB-SLAM3 and DVO SLAM are as follows:
>
> | ATE[cm]      | f3/wk_xyz | f3/wk_half|  f3/wk_static |  f3/st_half| Avg.
> | :---        |    :----:   |   :----:   |     :----:   | :----:   |  :----:   |
> |  DVO SLAM      |  59.7   |     52.9   |   21.2  | 6.2 |  35.0 |
> |  ORB-SLAM3    |   28.1  | 30.5   | 2.1  | 2.6 | 15.9 |
> |  Ours | 8.3  | 5.6 | 1.7  | 4.4 | 5.0 |
>
> **Q4: Degenerate case for computing the fundamental matrix**.
> Actually, the issue of degeneracy in fundamental matrix computation is a fundamental problem in multi-view geometry and should be investigated as a standalone problem. This is not the contribution of our paper. We employ an engineering trick to address this problem, similar to approaches used in ORB-SLAM2 and ORB-SLAM3. We compute the fundamental matrix $\mathbf{F}$ and homography matrix $\mathbf{H}$, utilizing the RANSAC method and reprojection error to score the computed matrices $SF$ and $SH$. Then, we determine the ratio of scores with the equation $rh = SH/(SH+SF)$. If the value of $rh$ is bigger than 0.5, we choose the homography matrix for the pose estimation; otherwise, we choose the fundamental matrix.
>
> > [1] Emanuele Palazzolo, Jens Behley, Philipp Lottes, Philippe Giguere, and Cyrill Stachniss. Refusion: 3d reconstruction in dynamic environments for rgb-d cameras exploiting residuals. In IROS, 2019.
>
> > [2] Berta Bescos, Jose M Facil, Javier Civera, and Jose Neira. Dynaslam: Tracking, mapping, and inpainting in dynamic scenes. RAL, 2018.
>
> > [3] Binbin Xu, Wenbin Li, Dimos Tzoumanikas, Michael Bloesch, Andrew Davison, and Stefan Leutenegger. Mid-fusion: Octree-based object-level multi-instance dynamic slam. In ICRA, 2019.
>
> > [4] Carlos Campos, Richard Elvira, Juan J Gomez Rodrıguez, Jose MM Montiel, and Juan D Tardos. Orb-slam3: An accurate open-source library for visual, visual–inertial, and multimap slam. IEEE TRO, 2021.
>
> > [5] Christian Kerl, J ̈urgen Sturm, and Daniel Cremers. Dense visual slam for rgb-d cameras. In IROS, 2013.
>
> > [6] Raul Mur-Artal and Juan D Tardos. Orb-slam2: An open-source slam system for monocular, stereo, and rgb-d cameras. IEEE TRO, 2017.

---

> ### Author Response · Authors · 2023-11-23
> **Responce to reviewer HXpV (2/2)**
>
> **Q5: Computing resources and Computation times analysis**.
> We have reported the time consumption of the tracking and mapping process in Section 4.4. Please refer to Table 5 in the original paper. Our system outperforms E-SLAM [7] in terms of time consumption and maintains a comparable level with Co-SLAM [8]. Therefore, our system can run in real-time like Co-SLAM, which is reported as a real-time SLAM method in paper [8]. In terms of computing resources, we run our RoDyn-SLAM on an RTX 3090Ti GPU, taking roughly 4GB of memory in total (please see "Implementation detail" part in the Experiment section).
>
>
> **Contribution**.
> Please allow us to briefly review the contribution of our Rodyn-SLAM. We propose a robust dynamic RGB-D SLAM with neural implicit representation. Existing neural RGB-D SLAM methods rely on a static environment assumption and do not work robustly within a dynamic environment due to the inconsistent observation of geometry and photometry. To tackle this problem, we propose a dynamic object identification method that combines the optical flow and semantics prior to generating the motion mask and filters the invalid sample ray to recover the static scene map. To further improve the accuracy of pose estimation in dynamic scenes, we propose a divide-and-conquer pose optimization algorithm utilizing edge projection loss to enhance data association between convisible frames. Extensive experiments are conducted on the two challenging datasets, and the results show that RoDyn-SLAM achieves state-of-the-art performance among recent neural RGB-D methods in both accuracy and robustness.
>
> > [7] JMohammad Mahdi Johari, Camilla Carta, and Franc ̧ois Fleuret. Eslam: Efficient dense slam system based on hybrid representation of signed distance fields. In CVPR, 2023.
>
> > [8] Hengyi Wang, Jingwen Wang, and Lourdes Agapito. Co-slam: Joint coordinate and sparse parametric encodings for neural real-time slam. In CVPR, 2023.

---

### Author Response · Authors · 2023-11-22
**Response to all reviewers**

We thank all the reviewers for their insightful comments. We have revised the paper as suggested by the reviewers, and summarized the major changes as follows:

* We have revised the introduction to motivate the benefits compared to existing dynamic dense RGB-D SLAM methods.

* We add Table 9 to evaluate the pose estimation results compared with the current SOTA neural RGB-D SLAM methods in static scenes. Please refer to Appendix A.5 for more details.

* We add the comparison results of MID-Fusion and DROID-SLAM in Table 2 and Table 3, respectively.

* The reconstruction evaluation of TSDF-fusion with gt pose on the BONN RGB-D dynamic dataset is added in Appendix A.4. Please refer to Table 8.

* A video demo on real-world datasets (TUM and BONN) is added to supplementary material.

* We adjust some statements and correct the typos in the main text.

* We highlight the text modified in the revised version.

The other concerns raised by the reviewers have also been addressed individually.

---

### Meta-Review · Area_Chair_h9et · 2023-12-07

**Metareview:**

A dynamic SLAM framework with neural radiance field is proposed, where a motion mask generation method is introduced to filter out the invalid sampled rays. To further improve the accuracy of pose estimation, a divide-and-conquer pose optimization algorithm is designed that distinguishes between keyframes and non-keyframes. Experiments show the effectiveness of the proposed method.

The paper received two “marginally below the acceptance threshold” ratings, one “accept” rating, and one “marginally above the acceptance threshold” rating.

Reviewer HXpV thinks there are only two datasets used for testing. Reviewer HXpV cares about the degenerate cases, the comparisons, and the computation times. In rebuttals, the authors think the used datasets are sufficient to evaluate the performance of dynamic SLAM system.
However, I would like to say the two used datasets are both indoor. A popular outdoor dataset KITTI isn’t used. Is the method only suitable to indoor scenes? Why can’t it be used outdoor scenes? There no descriptions on this problem. Moreover, the added experiments were performed to compare with ORB-SLAM, DVO-SLAM, and TSDF-Fusion. But the details are not given clear. Did all of them remove dynamic points? For the traditional methods, we can also detect dynamic points or outliers by simple point matching methods and then compute the camera pose. Did you remove them? Moreover, the neural methods use prior data but the traditional methods don’t. The comparison fairness is not clear.

Reviewer az7R thinks the motivation is not entirely clear, the comparisons against static neural SLAM may not be entirely fair, TSDF fusion results are not reported in tables, and the paper's visualization results are impressive. The authors revised the paper for the motivation. However, the comparison fairness is not described clear. The rating of Reviewer az7R is not changed.

Reviewer x3wF thinks the substantial architectural changes or novelty for NeRF-based SLAM are not introduced. Reviewer x3wF has doubt: is the method competitive with other SLAM systems in static scenes? Reviewer x3wF also thinks the used GBA in loop closure is not an actual GBA. The authors gave their rebuttals. The added Table 9 shows the method is not competitive in static scenes. The authors agree with the GBA problem.

Based on the above comments, the paper needs further revisions and charities for some details. The decision was to reject the paper.

**Justification For Why Not Higher Score:**

A popular outdoor dataset KITTI isn’t used. Is the method only suitable to indoor scenes? Why can’t it be used outdoor scenes? There no descriptions on this problem. Moreover, the added experiments were performed to compare with ORB-SLAM, DVO-SLAM, and TSDF-Fusion. But the details are not given clear. Did all of them remove dynamic points? For the traditional methods, we can also detect dynamic points or outliers by simple point matching methods and then compute the camera pose. Did you remove them? Moreover, the neural methods use prior data but the traditional methods don’t. The comparison fairness is not clear.

Reviewer az7R thinks the motivation is not entirely clear, the comparisons against static neural SLAM may not be entirely fair, TSDF fusion results are not reported in tables, and the paper's visualization results are impressive. The authors revised the paper for the motivation. However, the comparison fairness is not described clear. The rating of Reviewer az7R is not changed.

Reviewer x3wF thinks the substantial architectural changes or novelty for NeRF-based SLAM are not introduced. Reviewer x3wF has doubt: is the method competitive with other SLAM systems in static scenes? Reviewer x3wF also thinks the used GBA in loop closure is not an actual GBA. The authors gave their rebuttals. The added Table 9 shows the method is not competitive in static scenes. The authors agree with the GBA problem.

The paper needs further revisions and charities for some details.

**Justification For Why Not Lower Score:**

N/A

---

### Decision · Program_Chairs · 2024-01-16

Reject